# TopBP1 biomolecular condensates as a new therapeutic target in advanced-stage colorectal cancer

Laura Morano[1], Nadia Vezzio-Vié[2], Adam Aissanou[2], Tom Egger[1], Antoine Aze[1], Solène Fiachetti[1], Benoit Bordignon[3], Cedric Hassen-khodja[3], Hervé Seitz[1], Louis-Antoine Milazzo[2], Véronique Garambois[2], Laurent Chaloin[4], Nathalie Bonnefoy[2], Céline Gongora[2]*, Angelos Constantinou[1], Jihane Basbous[1]*

[1]Institut de Génétique Humaine, Université de Montpellier, CNRS, Montpellier, France; [2]IRCM, Université de Montpellier, ICM, INSERM, Montpellier, France; [3]Montpellier Ressources Imagerie, BioCampus, University of Montpellier, CNRS, INSERM, Montpellier, France; [4]Institut de Recherche en Infectiologie de Montpellier (IRIM), Université de Montpellier, CNRS, Montpellier, France

## eLife Assessment

This **valuable** study reveals that the GSK-3 inhibitor AZD2858 inhibits the formation of TOPBP1 condensates and hence DNA damage responses in colorectal cancer cells. The evidence supporting the claims of the authors is **convincing**, although uncovering how this drug blocks bio-condensate formation would have strengthened the study. The work will be of interest to cancer researchers searching for synergistic drug combination strategies.
[Editors' note: this paper was reviewed by Review Commons.]

*For correspondence:
celine.gongora@inserm.fr (CG);
jihane.basbous@igh.cnrs.fr (JB)

Competing interest: The authors declare that no competing interests exist.

**Abstract** In cancer cells, ATR signaling is crucial to tolerate the intrinsically high damage levels that normally block replication fork progression. Assembly of TopBP1, a multifunctional scaffolding protein, into condensates is required to amplify ATR kinase activity to the levels needed to coordinate the DNA damage response and manage DNA replication stress. Many ATR inhibitors are tested for cancer treatment in clinical trials, but their overall effectiveness is often compromised by the emergence of resistance and toxicities. In this proof-of-concept study, we propose to disrupt the ATR pathway by targeting TopBP1 condensation. First, we screened a molecule-based library using a previously developed optogenetic approach and identified several TopBP1 condensation inhibitors. Among them, AZD2858 disrupted TopBP1 assembly induced by the clinically relevant topoisomerase I inhibitor SN-38, thereby inhibiting the ATR/Chk1 signaling pathway. We found that AZD2858 exerted its effects by disrupting TopBP1 self-interaction and binding to ATR in mammalian cells, and by increasing its chromatin recruitment in cell-free *Xenopus laevis* egg extracts. Moreover, AZD2858 prevented S-phase checkpoint induction by SN-38, leading to increased DNA damage and apoptosis in a colorectal cancer cell line. Lastly, AZD2858 showed a synergistic effect in combination with the FOLFIRI chemotherapy regimen in a spheroid model of colorectal cancer.

## Introduction

In every cell, thousands of DNA lesions are induced by endogenous and environmental agents (*Lindahl and Barnes, 2000*). To overcome their deleterious effects, cells have evolved an intricate network of DNA damage response pathways that detect, signal, and repair DNA lesions. These pathways work in

coordination with key physiological processes, such as cell cycle progression, chromatin remodeling, and transcription (*Ciccia and Elledge, 2010*; *Jackson and Bartek, 2009*). Cancer cells typically exhibit higher levels of genotoxic stress than non-transformed cells due to their altered metabolism and high proliferation rate (*Jackson and Helleday, 2016*; *Luo et al., 2009*; *Macheret and Halazonetis, 2015*). As a complement to radiotherapy and chemotherapy that overload cancer cells with DNA lesions, the aim of novel anticancer strategies is to increase the sensitivity of cancer cells to genotoxic stress by targeting DNA damage signaling and DNA repair mechanisms (*Jackson and Helleday, 2016*; *Brown et al., 2017*).

For example, ATR signaling activates cell cycle checkpoints (*Brown and Baltimore, 2000*; *Eykelenboom et al., 2013*; *de Klein et al., 2000*), promotes DNA repair (*Tercero and Diffley, 2001*), regulates the firing of replication origins (*Toledo et al., 2013*), and the cellular pool of deoxyribonucleotides (*D'Angiolella et al., 2012*; *Le et al., 2017*; *Lopez-Contreras et al., 2015*; *Zhang et al., 2009*). The ATR signaling pathway is also essential for the proliferation of cancer cells that display intrinsically higher levels of genotoxic stress. Therefore, several clinical trials are evaluating the efficacy of ATR inhibitors in patients with cancer (*Lecona and Fernandez-Capetillo, 2018*), particularly cancers in which the oncogene CDC25A is overexpressed (*Ruiz et al., 2016*). However, as kinase inhibitors often exert a strong selective pressure for the acquisition of drug resistance through kinase mutations, alternative therapeutic options must be identified (*Bhullar et al., 2018*).

Unlike classical approaches that target the activity of a specific protein with small molecule inhibitors, recent cell biology advances suggest that targeting biomolecular condensates may provide conceptually novel approaches to drug discovery (*Mitrea et al., 2022*). Biomolecular condensates are subcellular compartments that selectively concentrate hundreds of proteins and nucleic acids, without a surrounding membrane (*Banani et al., 2017*; *Mittag and Pappu, 2022*). These structures underlie the spatiotemporal organization of multiple biological processes. Moreover, the aberrant properties of condensates have been implicated in many diseases, including neurodegeneration, cardiomyopathy, cancer, and viral infections (*Mitrea et al., 2022*). Condensate formation is typically driven by multivalent scaffold proteins that function as central nodes in molecular networks (*Banani et al., 2017*; *Mittag and Pappu, 2022*). Therefore, one potential approach to disturb the condensate properties and functions is to interfere with their essential protein-protein or protein-nucleic acid interactions.

In humans, the main ATR activator in S-phase is Topoisomerase IIβ binding protein 1 (TopBP1), a scaffold protein that includes nine BRCA1 carboxyl terminal (BRCT) protein-protein interaction motifs (*Kumagai et al., 2006*; *Wardlaw et al., 2014*) and an intrinsically disordered ATR activation domain located between BRCT6 and BRCT7-8 (*Kumagai et al., 2006*; *Zhou et al., 2013*). Recent studies indicate that TopBP1 activates the ATR signaling pathway via the formation of biomolecular condensates, which appear as nuclear foci by immunofluorescence microscopy (*Frattini et al., 2021*). The available evidence suggests that ATR activation is a multistep process (*Koundrioukoff et al., 2013*; *Egger et al., 2024*). TopBP1 is recruited to ATR-activating DNA structures by MRE11 (*Duursma et al., 2013*) and binds to RPA (*Acevedo et al., 2016*), ATRIP (*Mordes et al., 2008*), and the 9-1-1 complex (*Delacroix et al., 2007*; *Lee et al., 2007*) to form a stable ATR-activating complex. Then, TopBP1 is phosphorylated by the basal kinase activity of ATR, triggering the assembly of TopBP1 condensates (*Frattini et al., 2021*). We found that TopBP1 condensation functions as a molecular switch that amplifies ATR activity up to the level required for signal transduction by the checkpoint effector kinase Chk1 (*Frattini et al., 2021*).

Here, we explored the feasibility of targeting TopBP1 condensates as a novel strategy to interfere with ATR signaling and sensitize colorectal cancer (CRC) cells to chemotherapeutic drugs. We present a high-throughput optogenetic screening approach to identify small-molecule modulators of TopBP1 condensation. Using this approach, we identified AZD2858, a known glycogen synthase 3β (GSK-3β) inhibitor, as a potential modulator of TopBP1 condensation. We demonstrated in a CRC cell line that AZD2858 (at very low doses) inhibits the assembly of endogenous TopBP1 and suppresses the activation of the ATR/Chk1 pathway induced by the topoisomerase I inhibitor SN-38, the active metabolite of irinotecan. Furthermore, the combination of AZD2858 with FOLFIRI (5-fluorouracil+SN-38) displayed a synergistic effect in CRC cell spheroid cultures, including in spheroids from CRC cells resistant to SN-38. Incubation with AZD2858 for 6–12 hr improved the induction of apoptosis by SN-38 and FOLFIRI by dampening the S checkpoint. Our observations suggest that the cytotoxic effects observed with the AZD2858 and SN-38 combination are mainly explained by disruption of TopBP1

assembly, rather than by the direct involvement of the GSK-3β pathway. These data support the feasibility of targeting condensates formed in response to DNA damage to improve chemotherapy-based cancer treatments.

## Results

### Identifying modulators of TopBP1 condensation with an optogenetic approach

Resistance to conventional chemotherapies is the major cause of therapy failure in patients with cancer. In CRC, irinotecan resistance is often caused by overexpression of efflux pumps (*Wu et al., 2020*). However, additional pathways may also contribute, particularly the DNA damage response pathways that are frequently activated in various cancer types to cope with increased DNA replication stress. TopBP1 has been implicated in oxaliplatin resistance in gastric cancer (*Fang et al., 2022*) and is overexpressed in several cancer types (*Forma et al., 2013*). We first examined the number of TopBP1 condensates in the CRC cell lines HCT116 and HCT116-SN50 (50-fold more resistant to SN-38, the active metabolite of irinotecan, than the parental HCT116 cell line) (*Candeil et al., 2004*) and observed a significantly higher number of TopBP1 foci in HCT116-SN50 cells (*Figure 1—figure supplement 1*). Moreover, our previous findings suggest that TopBP1 condensation is required to amplify ATR activity (*Frattini et al., 2021*). Therefore, targeting TopBP1 assembly could be a promising strategy for cancer treatment. To this end, we developed an optogenetic system for screening molecules that modulate the formation of TopBP1 condensates. We used the stable Flp-In 293 T-Rex cell line in which expression of TopBP1 fused to the photoreceptor Cry2 and mCherry (optoTopBP1) is induced by doxycycline. Cry2 forms tetramers when exposed to 488 nm light and thereby nucleates the assembly of TopBP1 condensates (*Frattini et al., 2021*). The advantage of this optogenetic approach is that TopBP1 condensates are formed on demand, within minutes, in the absence of DNA damaging agents that can induce confounding effects after prolonged treatment. After a 2 hr incubation step of optoTopBP1-expressing Flp-In 293 T-Rex cells with 10 μM of each molecule from the TargetMol library of preclinical and clinical compounds, we exposed cells to an array of blue light-emitting diodes for 5 min of light-dark cycles (4 s light followed by 10 s dark) to induce opto-TopBP1 condensation (*Figure 1A*). After quantification of the number of TopBP1 foci per nucleus, we assigned a z-score to each candidate molecule based on this count. We considered molecules with a z-score lower than –2 as potential TopBP1 condensation inhibitors (red dots in *Figure 1B*). We identified >300 molecules that could be potential TopBP1 condensation inhibitors among the 4730 compounds. Then, we selected 131 compounds (based on the best z-scores) and assessed their effect on the ATR pathway activation in HCT116 cells using two complementary approaches. First, we evaluated these compounds at two low concentrations (1 and 10 μM) using a quantitative immunofluorescence assay (*Vezzio-Vié et al., 2022*) in flat-bottomed 384-well plates to monitor phosphorylation at S315 of the kinase effector Chk1 following incubation with SN-38 alone or with each of the 131 preselected compounds (*Figure 1C*). This allowed us to select the 10 most effective drugs (MCB-613, OTSSP167, AZD2858, P005091, tubercidin, dactolisib, physalin F, PFK158, PF562271, torkinib) based on their ability to inhibit Chk1 phosphorylation. For the second approach, we incubated HCT116 cells with 5-fluorouracil (5-FU) and SN-38 (FOLFIRI thereafter) to mimic in vitro the first-line chemotherapy regimen for patients with metastatic CRC (i.e. FOLFIRI: folic acid or leucovorin, 5-FU, and irinotecan) and each of the 10 drugs and detected their interactions using a full-range dose matrix approach and SRB cytotoxicity assays. We quantified cell survival and the additive, synergistic, or antagonistic effects. We observed the strongest synergistic interactions between FOLFIRI and AZD2858 (a known GSK-3β inhibitor). Consequently, we focused our study on AZD2858, which was efficient already at 1 μM (*Figure 1D*).

### AZD2858 hampers SN-38-induced endogenous TopBP1 condensate formation and ATR activation

We then evaluated AZD2858 capacity to modulate endogenous TopBP1 condensation in HCT116 cells. To this aim, we induced TopBP1 condensation by incubating cells with 300 nM SN-38 and/or 100 nM AZD2858 for 2 hr. Co-incubation with AZD2858 resulted in a significant reduction in the number of endogenous TopBP1 condensates per nucleus (*Figure 2A*). Co-incubation with AZD2858

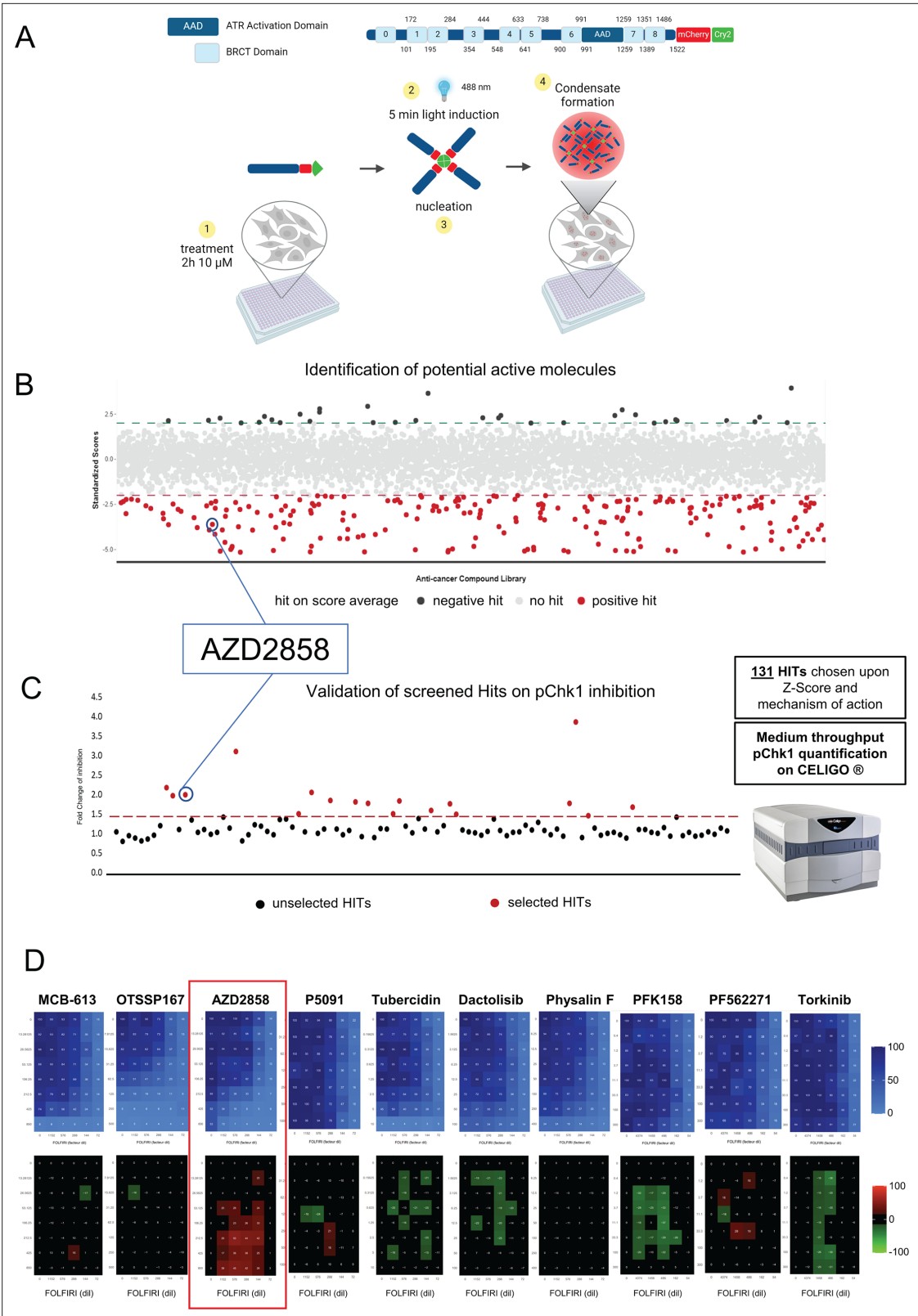

**Figure 1.** Identification of TopBP1 condensation inhibitors by high-throughput screening. (**A**) Schematic description of the high-throughput screening system. Doxycycline-inducible TopBP1 was fused to mCherry and the light-sensitive cryptochrome 2 (Cry2) at its C-terminus and stably integrated in Flp-In 293 T-Rex cells. A 5 min blue light exposure (cycles of 4 s on and 10 s off) allows inducing optoTopBP1 condensation in the absence of DNA damage. Cells were grown in 384-well plates, incubated with 10 µM TargetMol molecules for 2 hr before inducting optoTopBP1 condensate formation.

*Figure 1 continued on next page*

*Figure 1 continued*

(**B**) Graphical representation of the screening results. Drugs with a z-score lower than –2 were considered inhibitors, and drugs with a z-score higher than 2 were considered activators of TopBP1 condensate formation. The screening was performed in triplicate, and each dot represents the mean z-score of a molecule. (**C**) Graphical representation of the screening confirmation in HCT116 cells. Chk1 phosphorylation at S345 was assessed using Celigo immunofluorescence imaging after 2 hr co-incubation with SN-38 and each of the 131 drugs selected from the first screening. Drugs leading to Chk1 phosphorylation (pChk1) inhibition (>1.5-fold change) were considered promising candidates, and the 10 molecules leading to the highest inhibition were selected for the next screening. (**D**) Viability and synergy matrices obtained after HCT116 cell incubation with increasing concentrations of FOLFIRI (5FU from 0.009 to 0.148 µM, SN38 from 0.077 to 1.235 nM) and each of the 10 drugs (MCB-613 13.3–850 nM, OTSSP167 7.8–500 nM, AZD2858 13.3–850 nM, P005091 31.5–1000 nM, tubercidin 0.15–10 nM, dactolisib 1.58–100 nM, physalin F 6.25–400 nM, PFK158 1.2–300 nM, PF562271 1.2–300 nM, torkinib 0.4–300 nM) for 96 hr. Cell viability was assessed with the SRB assay. Blue matrices represent cell viability. The black/red/green matrices represent additivity/synergy/antagonism, respectively. The synergy matrices were calculated with an R script (see Materials and methods).

The online version of this article includes the following figure supplement(s) for figure 1:

**Figure supplement 1.** TopBP1 forms more foci in SN-38-resistant HCT116 cells.

also led to a decrease in the number of 53BP1 foci (*Figure 2—figure supplement 1A*). 53BP1 forms biomolecular condensates (*Kilic et al., 2019*) and might interact with TopBP1 (*Cescutti et al., 2010*; *Bigot et al., 2019*). Conversely, co-incubation with AZD2858 did not modulate the number of PML nuclear bodies, which are membrane-less organelles unlike the Arsenic treatment, which is known to induce the formation of PML nuclear bodies (*Jaffray et al., 2023*; *Figure 2—figure supplement 1B*). We then evaluated the effect of incubation of HCT116 cells with AZD2858 and/or SN-38 for 2 hr on ATR signaling (TopBP1 target), as well as on ATM (Ataxia Telangiectasia Mutated), another phosphoinositide 3-kinase-related protein kinase. Western blot analysis showed that SN-38-mediated ATR signaling induction was severely reduced by co-incubation with 100 nM AZD2858, as indicated by the decrease in Chk1 phosphorylated at S345 and RPA32 phosphorylated at S33 (*Figure 2B*). We obtained a similar result for Chk1 phosphorylated at S345 using the Celigo cytometer in cells incubated with AZD2858 and FOLFIRI (*Figure 2C*). Co-incubation of AZD2858 with SN-38 or FOLFIRI did not affect the ATM pathway, as indicated by the unchanged levels of ATM phosphorylated at S1981 and Chk2 phosphorylated at T68 (*Figure 2B* and *Figure 2—figure supplement 2A*). The level of γH2AX, a marker of DNA damage, was not affected by co-incubation with AZD2858 and SN-38 (or FOLFIRI) compared with SN-38 (or FOLFIRI) alone (*Figure 2B and C*). We obtained similar results (i.e. decrease in Chk1 phosphorylated at S345 and unchanged γH2AX levels) after co-incubation with FOLFIRI+AZD2858 for 20 hr (*Figure 2—figure supplement 2B*). Altogether, these findings suggest that AZD2858 significantly affects the ATR/Chk1 signaling pathway by inhibiting the assembly of TopBP1 condensates induced by the DNA-damaging agent SN-38.

## AZD2858 modulates ATR/Chk1 signaling by disrupting TopBP1 assembly

As AZD2858 is a known GSK-3β inhibitor, we asked whether its effects on the ATR/Chk1 signaling pathway might be mediated by inhibiting the GSK-3β pathway. We incubated SW620 cells in which endogenous GSK-3β was depleted or not by shRNA with SN-38 and/or AZD2858 (0.1 and 1 µM) for 2 hr. SN-38-induced Chk1 phosphorylation at S345 was comparable in GSK3-β-depleted cells (shGSK3) and control cells (shLuc) (*Figure 2—figure supplement 3A*). Moreover, the SN-38 and AZD2858 combination inhibited Chk1 phosphorylation in shGSK3 and shLuc cells, indicating that GSK-3β is not involved in Chk1 phosphorylation inhibition by AZD2858. The combination did not increase significantly the level of GSK-3β phosphorylated at S9, a marker of kinase inhibition, unlike what was observed with insulin, a well-known regulator of GSK-3β activity (*Cross et al., 1995*; *Figure 2—figure supplement 3B*). In addition, among the molecules tested from the TargetMol library, other more effective and specific GSK-3β inhibitors, compared with AZD2858, did not show any effect on light-induced optoTopBP1 condensation (*Supplementary file 1*). These findings suggest that although AZD2858 is a GSK-3β inhibitor, its modulation of the ATR/Chk1 pathway is not due to this function. To identify the underlying mechanisms, we used a biotin proximity-labeling approach to assess TopBP1 interaction with cellular partners in the presence of AZD2858 and/or SN-38 (*Alghoul et al., 2021*). For this, we used our optoTopBP1 construct tagged at its N-terminus with TurboID, an optimized biotin ligase that allows protein biotinylation within minutes (*Branon et al., 2018*). We incubated Flp-In 293 T-Rex cells that stably express TurboID-tagged optoTopBP1 with AZD2858 and/or SN-38

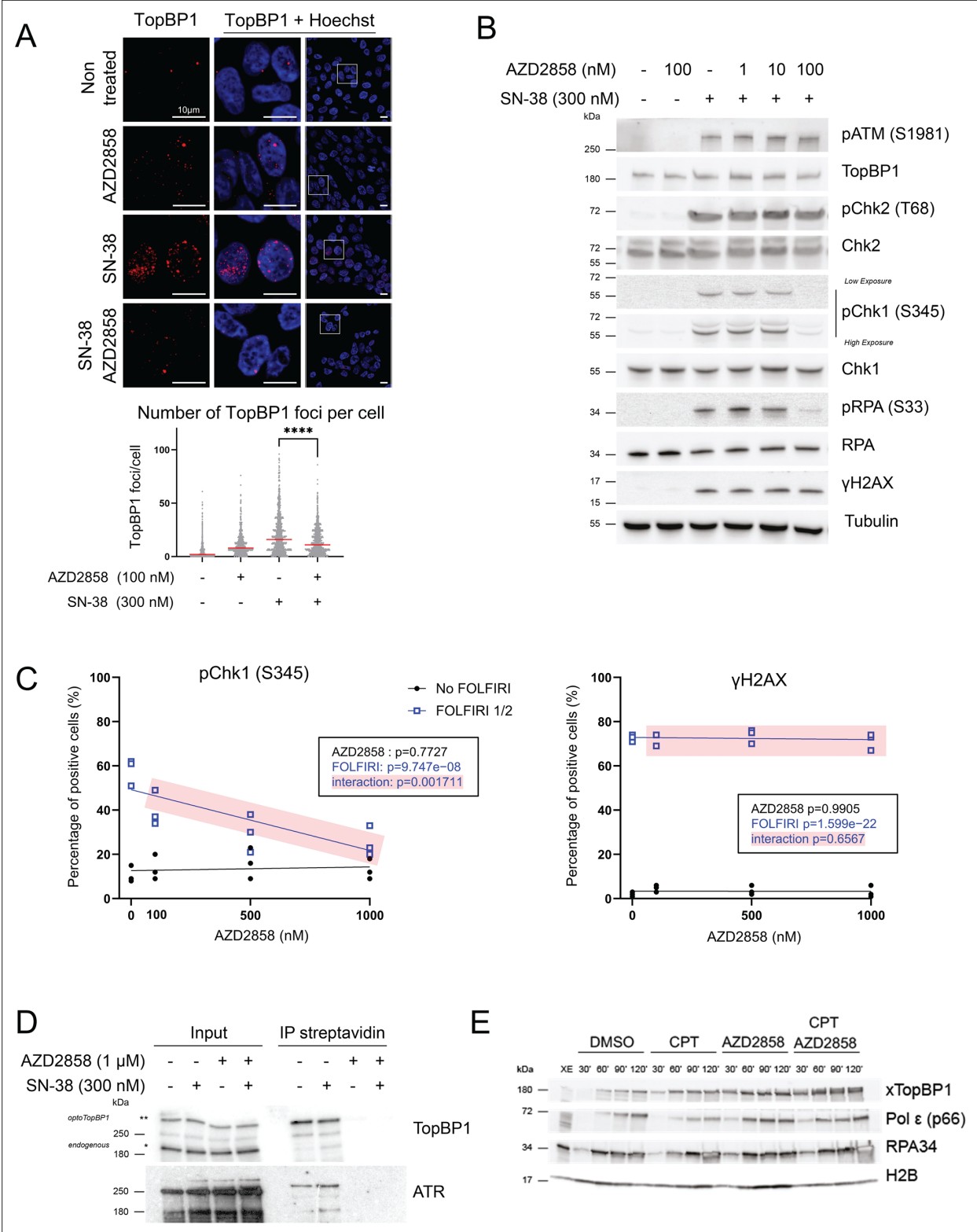

**Figure 2.** AZD2858 inhibits the formation of endogenous TopBP1 condensates and the ATR/Chk1 signaling pathway. (**A**) Representative immunofluorescence images (upper panels) of TopBP1 condensates in HCT116 cells incubated with AZD2858 (100 nM) and/or SN-38 (300 nM) for 2 hr, and the corresponding quantification (lower panel). Scale bars: 10 µm. The experiment was replicated three times. Cell profiler was used for quantifying TopBP1 foci (>1000 cells analyzed per condition). Generalized linear model analysis shows that SN38 and AZD2858 treatments exhibit significantly negative interaction on TopBP1 foci number (SN38 treatment p-value<2e-16; AZD2858 treatment p-value=0.1665; SN38:AZD2858

*Figure 2 continued on next page*

*Figure 2 continued*

interaction p-value<2e-16). For more details on the analyses, refer to the Materials and methods section. (**B**) Immunoblot of the indicated proteins after incubation of HCT116 cells with AZD2858 at the indicated concentrations and/or SN-38 (300 nM) for 2 hr. The experiment was replicated three times, and a representative replicate is shown. (**C**) Percentage of HCT116 cells positive for phosphorylated Chk1 (S345) or γH2AX expression as a function of the AZD2858 concentration (0 nM, 100 nM, 500 nM, 1000 nM) in the presence (blue) or not (black) of FOLFIRI (dilution: 1/2, corresponding to 6 µM of 5-fluorouracil [5-FU] and 50 nM of SN-38). Cells were incubated for 2 hr, and positive cells were identified using Celigo immunofluorescence imaging. The experiment was replicated three times, and each point represents a biological replicate. The statistical significance was determined by linear modeling interrogating the effect of each drug separately and their interaction (p-values given in the inset). (**D**) Immunoblotting of TopBP1 and ATR isolated with streptavidin beads from optoTopBP1-expressing cells incubated with doxycycline for 16 hr to induce optoTopBP1 expression, and also with AZD2858 (1 µM) and/or SN-38 (300 nM) for the last 2 hr. Biotin was added to the medium in all conditions for the last 30 min. Bands that correspond to endogenous and optoTopBP1 proteins are indicated with one and two stars (* and **), respectively. The experiment was replicated twice, and a representative result is shown. (**E**) Chromatin was extracted at the indicated time points (min) following the assembly of nuclei from sperm DNA incubated in *X. laevis* egg extracts incubated with DMSO (control), camptothecin (CPT; 55 µM), and/or AZD2858 (1 µM). Chromatin samples were analyzed by western blotting.

The online version of this article includes the following source data and figure supplement(s) for figure 2:

**Source data 1.** Original membranes corresponding to *Figure 2B, D and E* with labels.

**Source data 2.** Original membranes corresponding to *Figure 2B, D and E*.

**Source data 3.** Original data corresponding to *Figure 2C*.

**Figure supplement 1.** Impact of the AZD2858 and SN-38 combination on two nuclear biomolecular condensates.

**Figure supplement 2.** Effects of AZD2858 on components of the DNA damage response.

**Figure supplement 3.** AZD2858 effect on ATR signaling is not related to GSK-3β activity.

**Figure supplement 3—source data 1.** Original membranes corresponding to *Figure 2—figure supplement 3A and B* with labels.

**Figure supplement 3—source data 2.** Original membranes corresponding to *Figure 2—figure supplement 3A and B*.

for 2 hr and added biotin for the last 30 min. Isolation of biotinylated proteins using streptavidin-coated beads highlighted a drastic reduction in TopBP1 proximity with recombinant and endogenous TopBP1 and also with ATR upon incubation with AZD2858 alone or with SN-38 (*Figure 2D*). These findings suggest that AZD2858 disrupts TopBP1 self-association and its interaction with its primary activator ATR. This may contribute to the observed decrease in SN-38-induced TopBP1 foci following AZD2858 administration. Next, we determined whether TopBP1 recruitment to chromatin was altered by AZD2858 using the well-established cell-free DNA replication system derived from *X. laevis* egg extracts in which replication of demembranated sperm nuclei occurs synchronously, independently of transcription or protein synthesis. In the presence of the topoisomerase I inhibitor CPT, replication fork progression was challenged, leading to a reduction in polymerase (Pol) ε levels and an accumulation of DNA-bound TopBP1 and RPA to activate the checkpoint response (*Figure 2E*). When AZD2858 was added to the extract alone or with CPT, TopBP1 recruitment to chromatin was not inhibited. Our data demonstrate that combining CPT and AZD2858 earlier enhances the accumulation of replication-related factors (RPA, TopBP1, and Pol ε) on chromatin compared to CPT treatment alone, particularly visible at the 60 min after starting replication (*Figure 2E*). This may be due to increased origin activation, as normally seen when the S-phase checkpoint is inhibited (*Marheineke and Hyrien, 2004*). Collectively, these observations suggest that AZD2858 acts primarily on the ATR signaling pathway through TopBP1 self-association inhibition.

## AZD2858 inhibits induction of the S-phase checkpoint by SN-38

Chk1 plays a pivotal role as a mediator of the S-phase checkpoint (*Stracker et al., 2009*). As SN-38-induced phosphorylation of Chk1 was decreased after incubation with AZD2858, we investigated the impact of 2 hr treatments of SN-38 and/or AZD2858 on the cell cycle distribution using a BrdU/PI flow cytometry assay. BrdU incorporation was markedly reduced in SN-38-treated HCT116 cells, suggesting a slowdown of the S-phase (*Figure 3A*). However, when SN-38 was combined with AZD2858, this effect was partially reduced (red arrows in *Figure 3A*).

To determine whether this effect depends on replication fork progression or replication origin firing, we co-labeled replication tracks with two consecutive pulses of the thymidine analogs IdU and CldU (*Figure 3B*). As expected, CldU tracks were shorter in HCT116 cells incubated with SN-38 alone, while addition of AZD2858 significantly reduced the negative impact of SN-38 on replication fork

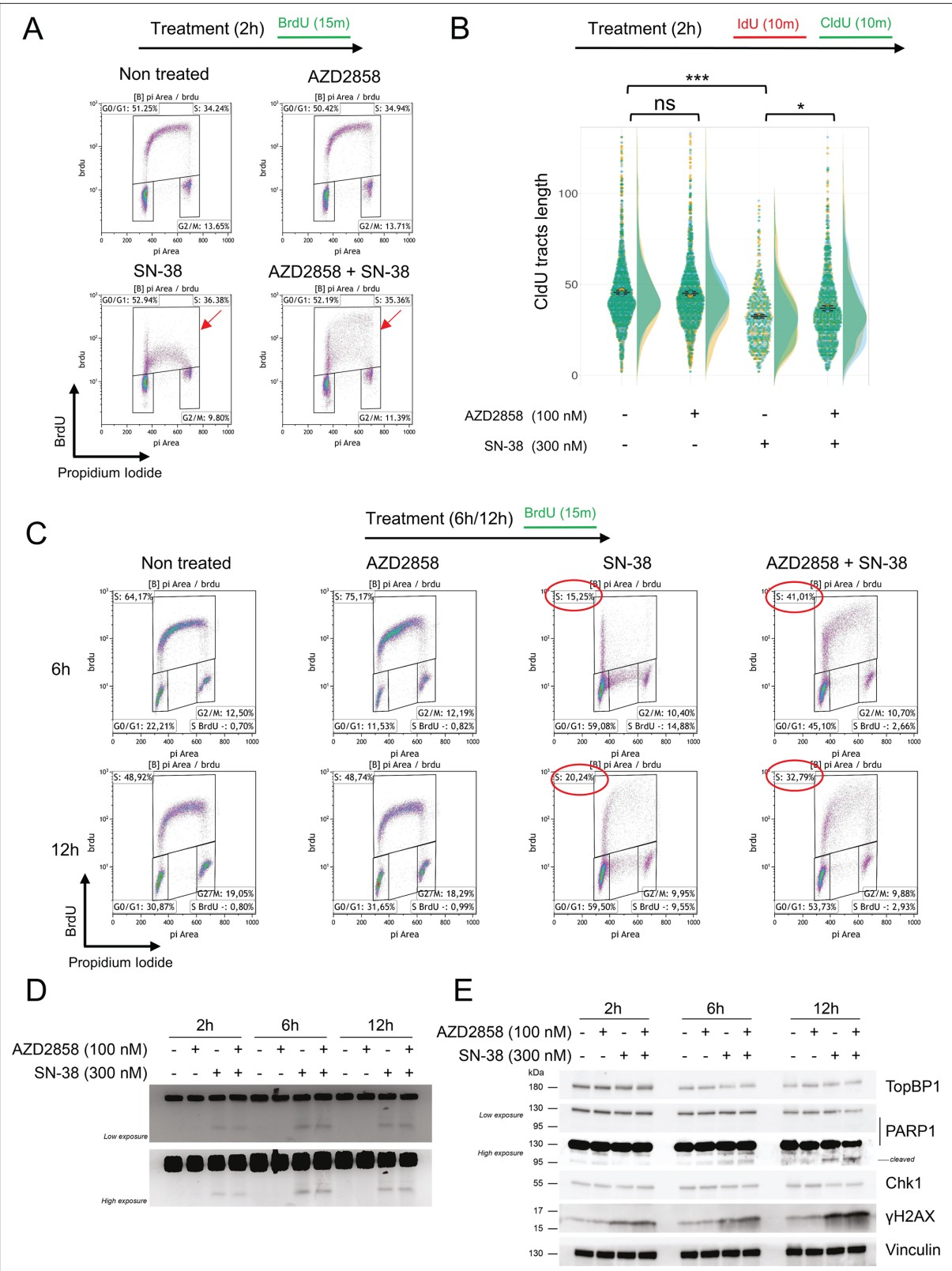

**Figure 3.** AZD258 inactivates the S-phase checkpoint in SN-38-treated cells. (**A**) Cell cycle analysis after a 2 hr incubation with AZD2858 (100 nM) and/or SN-38 (300 nM); 10 µM BrdU was added in the last 15 min of the 2 hr incubation. Cell debris was gated out, and BrdU incorporation was plotted against DNA content (stained with PI). Red arrows indicate BrdU incorporation. The gates for the S-phase population (BrdU-positive cells) were set broadly to prevent bias and ensure inclusion of cells with weak BrdU incorporation, particularly in the SN-38-only condition. The percentage of cells within this

*Figure 3 continued on next page*

*Figure 3 continued*

gate remains comparable across conditions, even though the FACS plot's overall shape changes, reflecting a shift in BrdU incorporation distribution and highlighting a heterogeneous population with varying levels of BrdU incorporation, especially in the combination treatment. 20,000 events per condition were analyzed. The $G_0/G_1$, S, and $G_2$/M gates are shown. The experiment was replicated three times, and a representative replicate is shown. More information is available in *Supplementary file 2*. (**B**) DNA fiber analysis of replication tracts labeled with two sequential 15 min pulses of IdU and CldU added at the end of the 2 hr incubation with AZD2858 (100 nM)±SN-38 (300 nM). The dot plots show the CldU tract length of individual replication forks. Data are pooled from n=3 biological replicates (yellow, blue, green). Data distribution is shown as half-violin plots. Circles represent the mean of each replicate, and the error bars represent the SEM of the means of the three replicates. Statistical significance was first assessed using ANOVA (p-value<2.2e-16). The t-test, represented in the figure, was then used to specifically compare the indicated conditions (ns: nonsignificant; *: p<0.05; ***: p<0.001). For more details on the analyses, refer to the Materials and methods section. (**C**) Same as (**A**) but after 6 hr or 12 hr of incubation with AZD2858 (100 nM) and/or SN-38 (300 nM). The gate of interest (S-phase) and the corresponding cell percentages are outlined in red. More information is available in *Supplementary file 3* and *Figure 3—figure supplement 1C and D*. (**D**) PFGE analysis of DNA damage in HCT116 cells incubated with AZD2858 (100 nM) and/or SN-38 (300 nM) for the indicated times. The experiment was replicated three times, and a representative replicate is shown. (**E**) Immunoblot of the indicated proteins after incubation with AZD2858 (100 nM) and/or SN-38 (300 nM) for the indicated times. The experiment was replicated three times, and a representative replicate is shown.

The online version of this article includes the following source data and figure supplement(s) for figure 3:

**Source data 1.** Original membranes corresponding to *Figure 3D and E* with labels.

**Source data 2.** Original membranes corresponding to *Figure 3D and E*.

**Source data 3.** Original data corresponding to *Figure 3B*.

**Figure supplement 1.** Cell cycle profiling of HCT116 cells incubated with AZD2858 and/or SN-38.

progression. We observe a heterogeneous population in the combination treatment, with some CldU tracts remaining short, similar to those in SN-38-treated cells, while others were longer, resembling those observed in untreated cells. This heterogeneity contributes to an overall increase in the mean length of CldU tracts in cells treated with the combination of AZD2858 and SN-38.

Moreover, the impact of the combination treatment becomes more pronounced over time. Indeed, there was a tendency for the SN-38-induced S-phase checkpoint to be weakened after 6 and 12 hr of co-treatment with AZD2858 and SN-38 (*Figure 3C*, *Figure 3—figure supplement 1C and D*). In the representative biological replicate shown, the percentage of BrdU incorporating cells increased from 15% (SN-38 alone) to 41% (combination) at 6 hr and from 21% (SN-38 alone) to 34% (combination) at 12 hr (*Figure 3C*, *Figure 3—figure supplement 1C and D*). These results indicate that the combination of AZD2858 and SN-38 affects replication fork progression and disrupts the cell cycle homeostasis, likely by inhibiting the ATR/Chk1 signaling pathway.

To further investigate the impact of AZD2858 on cell cycle progression, we performed a pulse-chase experiment in HCT116 cells. Cells were pulse-labeled with BrdU for 15 min, washed, and then incubated with SN-38 and/or AZD2858 for 6 and 12 hr. In the combination treatment, at 12 hr, we observed an increase in the proportion of cells transitioning into S-phase, which were initially in G1 (BrdU-negative) and progressed to a mid-S DNA content state (between 2N and 4N) (green squares) (*Figure 3—figure supplement 1A and B*). This observation aligns with previous findings, indicating that the S-phase checkpoint is partially alleviated in the combination treatment compared to SN-38 alone. Of note, in untreated cells and those treated with AZD2858 alone, the S-phase gate represents a new round of DNA synthesis, as indicated by the progression from the new G1 phase at 6 hr to the initiation of a new S-phase at 12 hr. However, in cells treated with SN-38 or the combination of SN-38 and AZD2858, the same S-phase population observed at 6 hr persists at 12 hr, reflecting a marked slowing of cell cycle progression.

Finally, the combined treatment led to a progressive accumulation of BrdU-positive cells with 4N DNA content, corresponding to cells that completed S-phase (BrdU-positive) and progressed into G2 phase (4N DNA content) (red squares) (*Figure 3—figure supplement 1A and B*). This effect was more pronounced after 12 hr of treatment, with the proportion of BrdU-positive cells in G2 phase increasing from 4% in SN-38-treated cells to 11% in the combination treatment, as shown in the representative experiment in *Figure 3—figure supplement 1B*. While this trend was consistently observed across all experiments, the increase was modest and did not reach statistical significance (*Figure 3—figure supplement 1F*). This suggests that the combination of SN-38 and AZD2858 attenuates the S-phase checkpoint, thereby activating a compensatory post-replicative cell cycle checkpoint that arrests the cell cycle before mitosis.

Abrogation of the replication checkpoint allows cells to enter and progress into the S-phase with unrepaired DNA damage. We evaluated the level of DNA double-strand breaks (DSBs) using PFGE and monitored γH2AX and cleaved PARP1 expression by western blotting in cells exposed to SN-38 and AZD2838 at various time points. DSBs remained elevated after the combined treatment (*Figure 3D*). Additionally, γH2AX and cleaved PARP1 levels increased at later time points of incubation (*Figure 3E*). Altogether, these findings indicate that the AZD2858 and SN-38 combination disrupts TopBP1 assembly, most likely by inhibiting ATR kinase activity amplification, a critical process for inducing the S-phase checkpoint and regulating DNA damage repair.

## AZD2858 treatment synergizes with conventional chemotherapy drugs

In the absence of an effective S-phase checkpoint, DNA damage can lead to apoptosis and cell death. To assess this, we stained HCT116 cells with PI and quantified the sub-G1 population to measure cell death. After 48 hr incubation with sublethal doses of SN-38 and AZD2858, the sub-G1 cell population was increased by 40% compared with untreated cells, indicative of apoptotic cells (*Figure 4A*) and consistent with the early increase of cleaved PARP1 (*Figure 3E*). We obtained the same results in cells incubated with AZD2858 and FOLFIRI. Specifically, in the representative experiment shown in *Figure 4A*, the sub-G1 population increased from 19% after incubation with SN-38 alone to 61% after incubation with SN-38+AZD2858, and from 27% after incubation with FOLFIRI alone to 71% after incubation with FOLFIRI+AZD2858 (*Figure 4B*). Additionally, the combined treatments (AZD2858 with SN-38 or FOLFIRI) led to an increase of cleaved PARP1 and cleaved caspase-3 (western blotting, *Figure 4C*), and to the accumulation of DNA DSBs (PFGE, *Figure 4D*). Then, we performed an SRB assay in HCT116 cells incubated with the AZD2858 and SN-38 combination, as previously described, and visualized the viability (blue) and synergy (black/red) matrices. The AZD2858 and SN-38 combination, like the AZD2858 and FOLFIRI combination (*Figure 1D*), synergistically inhibited cell survival (*Figure 4E*).

Next, we assessed whether the combination affected cell death or proliferation. First, we determined whether AZD2858, SN-38, and FOLFIRI alone induced cytotoxic (cell death) or cytostatic (inhibition of cell proliferation) effects by PI/Hoechst labeling of cells incubated with increasing concentrations of each drug. The three drugs displayed mainly cytostatic effects (*Figure 5—figure supplement 1A*), as indicated by the absence of cell death (lower panels) even at doses that decreased cell proliferation (upper panels). Next, we stained with PI/Hoechst and performed SRB assays in HCT116 cells, CT-26 cells (murine CRC cell line), and CCD 841 CoN (untransformed colorectal cells) incubated with increasing concentrations of AZD2858 and FOLFIRI (matrix analysis). We found that the combination was cytostatic at low drug doses and became cytotoxic at higher doses (*Figure 5—figure supplement 1B and C*), especially in areas where we detected synergy (*Figure 5A* and *Figure 5—figure supplement 1D*). However, no synergistic effect was observed on CCD 841 CoN cells with the AZD2858 and FOLFIRI combination (*Figure 5—figure supplement 2*). These results indicate that when the AZD2858+FOLFIRI combination shows synergistic effects, this interaction is cytotoxic. We confirmed the synergy of the FOLFIRI+AZD2858 combination in three additional human CRC cell lines: HT29, SW620, and SW480 (*Figure 5C*). We also performed a matrix analysis in two SN-38-resistant cell lines: HCT116 SN-6 and HCT116 SN-50 (*Gongora et al., 2011*; *Figure 5D and F*). The FOLFIRI+AZD2858 combination was synergistic in both SN-38-resistant cell lines, indicating that this combination may alleviate drug resistance in CRC.

Lastly, we tested the FOLFIRI+AZD2858 combination (matrix analysis) in 3D cell cultures that are physiologically closer to tumor growth compared to 2D cultures and allow mimicking the complexity of solid tumors from a structural, biochemical, and biophysical point of view. We prepared spheroids of parental HCT116 cells, SN-38-resistant HCT116 SN-6 and HCT116 SN-50 cells (*Figure 5B, E, and G*), and murine CT-26 cells (*Figure 5—figure supplement 1E*). We again observed that the combination was synergistic in all tested cell lines, demonstrating remarkable efficacy with nearly complete elimination of spheroids.

## Discussion

TopBP1 assembly is required for the activation of the ATR signaling pathway, a critical step in the DNA damage response and repair. We exploited our ability to control TopBP1 condensation and

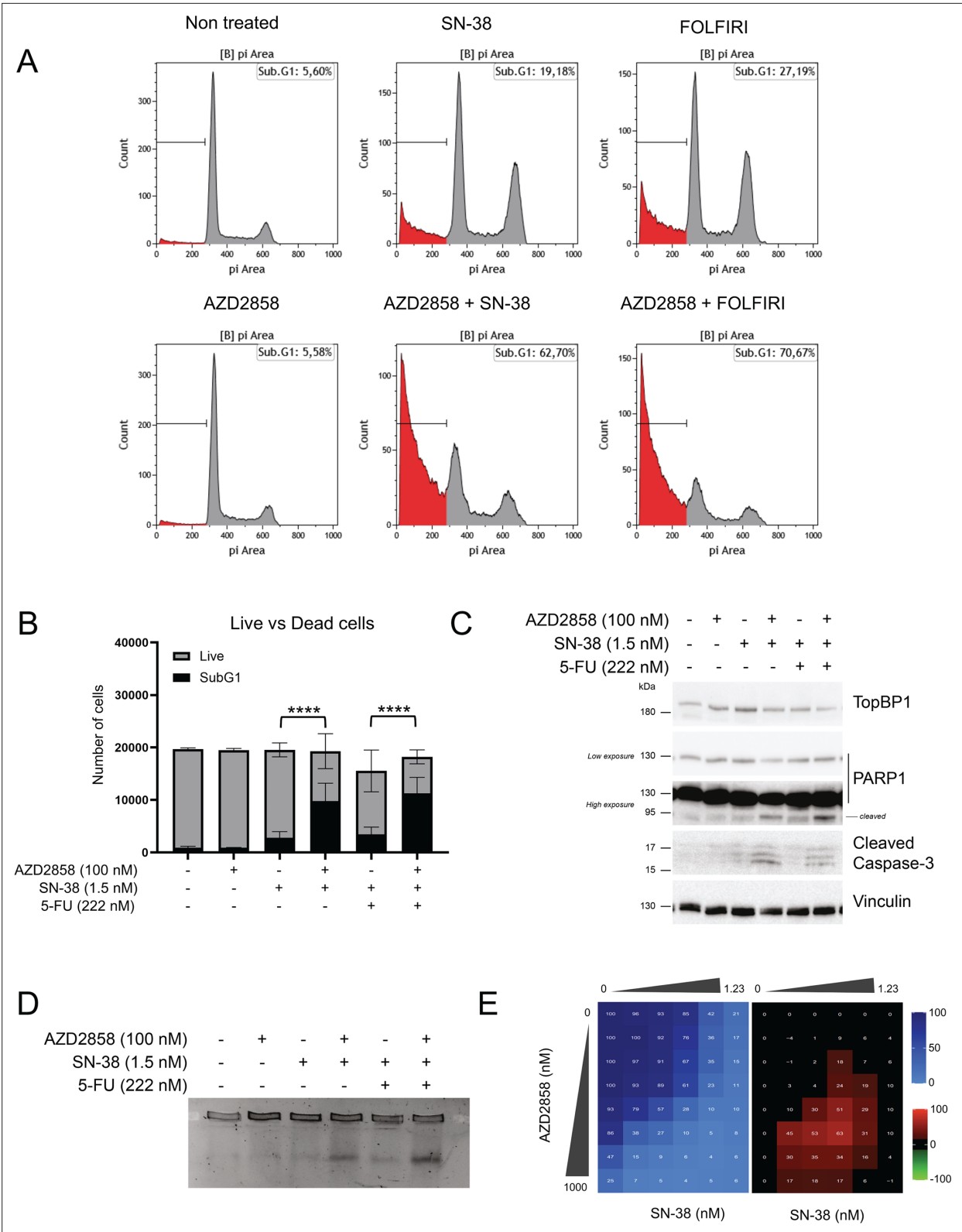

**Figure 4.** AZD2858 synergizes with SN-38 and FOLFIRI by inducing DNA damage and cell apoptosis. (**A**) Flow cytometry analysis to quantify sub-G1 HCT116 cells after 48 hr incubation with suboptimal doses of AZD2858 (100 nM) alone or with SN-38 (1.5 nM) or FOLFIRI (SN-38 1.5 nM; 5-FU 222 nM). The x-axis shows the DNA content (PI staining) and the y-axis the cell count. The presence of sub-G1 cells gated under the G1 peak suggests DNA fragmentation, a characteristic feature of apoptotic cell death. The experiment was replicated three times, and a representative replicate is shown.

*Figure 4 continued on next page*

*Figure 4 continued*

Minor gating adjustments were applied to best align the sub-G1 population with the DNA-content profiles, without affecting the statistical analysis shown in panel B. (**B**) Graph showing the number of sub-G1 cells (considered as dead cells) vs. the number of live cells, according to the indicated treatments (from A). Error bars represent the standard deviation from three individual biological replicates. The statistical significance was determined by linear regression analysis, and more information is available in *Supplementary file 4* (****: p<0.0001). (**C**) Immunoblot of the indicated proteins in HCT116 cells after 48 hr incubation as described in A. (**D**) PFGE analysis of DNA damage in HCT116 cells incubated for 48 hr as described in A. (**E**) HCT116 cells were incubated with increasing concentrations of SN-38 (0–1.23 nM) and AZD2858 (0–1000 nM) for 96 hr (2D culture system). Cell viability was assessed with the SRB assay. The synergy matrix was calculated as described in Materials and methods. The experiment was replicated three times, and a representative replicate is shown.

The online version of this article includes the following source data for figure 4:

**Source data 1.** Original membranes corresponding to *Figure 4C and D* with labels.

**Source data 2.** Original membranes corresponding to *Figure 4C and D*.

**Source data 3.** Original data corresponding to *Figure 4B*.

ATR activation using an optogenetic approach to develop a high-throughput screening procedure for identifying modulators of TopBP1 condensation. As proof of concept, we screened a TargetMol library of 4730 preclinical and clinical compounds. Through this approach, we found that AZD2858, a known GSK3-β inhibitor, is also a potential inhibitor of TopBP1 condensation. Using a complementary approach based on the irinotecan active metabolite SN-38 to induce DNA damage in CRC cells, we confirmed that AZD2858 disrupts endogenous TopBP1 condensates, thus affecting the ATR signaling pathway.

AZD2858 inhibits GSK-3β activity and activates the canonical Wnt/β-catenin signaling cascade (*Gao et al., 2017*). In addition, AZD2858 has cytotoxic effects in glioma cell lines through disruption of centrosome function and mitotic failure (*Brüning-Richardson et al., 2021*), and affects glioma proliferation and survival (*Gao et al., 2017*). Our findings show that AZD2858, at nanomolar concentrations, alters TopBP1 condensation without influencing GSK-3β activity and does not require the presence of GSK-3β to prevent Chk1 activation by SN-38. These observations clearly rule out the involvement of the GSK-3β pathway in TopBP1 condensate formation inhibition by AZD2858.

TopBP1 is overexpressed in several cancer types, including breast cancer (*Forma et al., 2012*) and advanced-stage CRC, as well as radiotherapy-resistant lung cancer cells and oxaliplatin-resistant gastric cancer (*Fang et al., 2022*). This overexpression is associated with high-grade tumors and poor prognosis. Notably, inhibition of TopBP1 expression increases the radiosensitivity of lung cancer and brain metastases (*Choi et al., 2014*) and also DNA damage in oxaliplatin-resistant gastric cancer cells (*Fang et al., 2022*). To date, no study assessed or visualized TopBP1 condensate levels in tumors, or investigated whether their presence is linked to poor prognosis. Here, we found that TopBP1 condensates are increased in SN-38-resistant CRC cells. Many attempts have been made to target TopBP1 activity. Weei-Chin Lin's team was the first to perform two large-scale molecular docking screens targeting the BRCT7/8 domains of TopBP1. They identified calcein AM and 5D4 as compounds that block TopBP1 oligomerization and demonstrated their anticancer activity in vivo (*Lin et al., 2023*; *Chowdhury et al., 2014*). Interestingly, 5D4 also disrupts TopBP1 interaction with E2F1, mutant p53, CIP2A, and MIZ1, leading to E2F1-mediated apoptosis, suppression of mutant p53, and repression of MYC activity (*Lin et al., 2023*).

We demonstrated that AZD2858 disrupts the interaction between TopBP1 and its canonical partner ATR and also its self-interaction. Additionally, in cell-free *X. laevis* egg extracts, AZD2858, alone or in combination with the topoisomerase I inhibitor CPT, enhanced the DNA binding occupancy of both TopBP1 and RPA32. These findings, combined with the cell cycle and DNA fiber results, suggest that AZD2858 abrogates the intra-S-phase and DNA elongation checkpoints induced by SN-38, leading to unscheduled DNA synthesis. In addition, incubation with SN-38+AZD2858 enhanced DNA damage and apoptosis, as indicated by (i) the increased γH2AX levels, (ii) the persistence of DSBs (PFGE assay), (iii) the expression of cleaved PARP1 and caspase-3 (apoptotic markers), and (iv) the accumulation of sub-G1 cells. Incubation with the AZD2858+SN-38 combination mimicked the effects observed with ATR inhibitors when combined with topoisomerase I inhibitors, such as CPT, topotecan, and SN-38 (*Chowdhury et al., 2014*; *Egger et al., 2022*). For instance, the ATR inhibitors VE-821 and VE-822 sensitize cancer cells to CPT derivatives by alleviating the intra-S-phase checkpoint (*Chowdhury et al., 2014*; *Egger et al., 2022*) and induce apoptosis mediated by caspase-3 when combined with SN-38

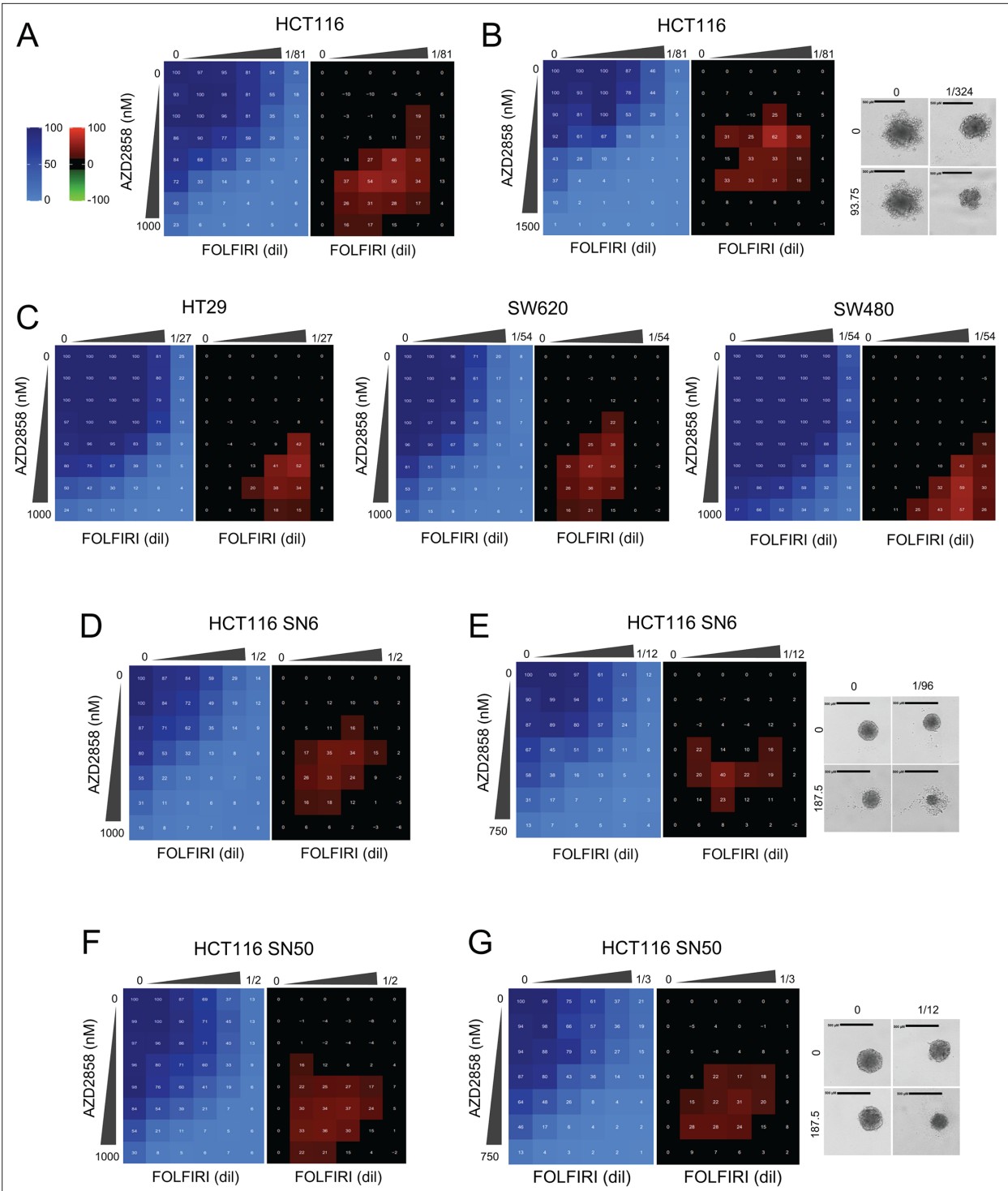

**Figure 5.** AZD2858 synergizes with FOLFIRI on a panel of colorectal cancer (CRC) cell lines, including HCT116 resistant to SN-38. (**A**) HCT116 cells were incubated with increasing concentrations of FOLFIRI (5-FU from 0.009 to 0.148 µM and SN-38 from 0.077 to 1.235 nM) and AZD2858 (from 15.6 to 1000 nM). Cell viability was assessed with the SRB assay in 2D cultures to generate the viability matrix (blue). The synergy matrices (black and red) were calculated as described in Materials and methods. (**B**) HCT116 cells were cultured in 3D to form spheroids and incubated with increasing concentrations of FOLFIRI (5-FU from 0.009 to 0.148 µM and SN-38 from 0.077 to 1.235 nM) and AZD2858 (from 23.4 to 1500 nM). Cell viability was assessed with CellTiter-Glo to obtain the viability matrix (blue). The synergy matrices (black and red) were calculated as described in Materials and methods. Representative bright-field images of spheroids are shown: untreated and after incubation with the drug concentrations giving the highest synergy score. (**C**) HT29, SW620, and SW480 CRC cells were incubated (2D culture) with increasing concentrations of FOLFIRI (HT29 cells: 5-FU from 0.03 to 1 µM

*Figure 5 continued on next page*

*Figure 5 continued*

and SN-38 from 0.07 to 2.35 nM; SW620 and SW480 cells: 5-FU: from 0.13 to 4.4 µM, SN-38: from 0.007 to 0.25 nM) and AZD2858 (from 15.6 to 1000 nM). (**D**) SN-38-resistant HCT116 SN-6 (six times more resistant to SN-38 than wild-type HCT116, 2D culture) were incubated with increasing concentrations of FOLFIRI (5-FU from 0.46 to 14.85 µM and SN-38 from 0.1 to 3.375 nM) and AZD2858 (from 15.6 to 1000 nM). (**E**) Same as in B but with SN-38-resistant HCT116 SN-6 cells. Drug concentrations were as follows: FOLFIRI (5-FU from 0.083 to 1.333 µM and SN-38 from 0.694 to 11.1 nM) and AZD2858 (from 46.8 to 750 nM). (**F**) Same as in (**D**) but with SN-38-resistant HCT116 SN-50 cells (50 times more resistant to SN-38 than wild-type HCT116). (**G**) Same as in B but with SN-38-resistant HCT116 SN-50 cells. Drug concentrations were as follows: FOLFIRI (5-FU from 0.75 to 12 µM and SN-38 from 6.25 to 100 nM) and AZD2858 (from 46.8 to 750 nM). Experiments were replicated three times, and representative replicates are shown.

The online version of this article includes the following figure supplement(s) for figure 5:

**Figure supplement 1.** The synergistic effect of the AZD2858+FOLFIRI combination is due to cytotoxicity.

**Figure supplement 2.** AZD2858 combined with FOLFIRI showed no synergistic effect on CCD 841 CoN cells.

because cells experience extensive DNA damage due to failure of the replication checkpoint (*Egger et al., 2022*). Importantly, the efficacy of several ATR inhibitors is currently being investigated in preclinical studies or in phase I/II clinical trials as monotherapy or in combination with chemotherapy-induced replication stress (*Qiu et al., 2018*). We also showed a synergistic effect of the AZD2858+-FOLFIRI combination, including in SN-38-resistant HCT116 cell lines, due to the increased apoptosis and accumulation of DNA damage. Given the similarity of mechanism between AZD2858 and ATR inhibitors, when combined with chemotherapy (SN-38 or CPT), we hypothesize that AZD2858 could prevent the development of resistance because its mechanism of action is not to inhibit the ATR kinase but to prevent the interaction of TopBP1 with ATR. FOLFIRI is used as a first-line chemotherapy regimen for patients with metastatic CRC and gastric/gastroesophageal cancer (*Van Cutsem et al., 2010*).

In the last decade, a major advance has been made in our understanding of the mechanisms and functions of various cytoplasmic and nuclear membrane-less organelles within cells, revolutionizing the field of cell biology. Recent studies highlighted the importance of compartmentalizing cellular components into mesoscale structures, known as biomolecular condensates, implicated in new physiological functions and in triggering the activation of specific signaling pathways. Many research projects have incorporated condensates into the understanding of cancer pathogenesis, leading to the development of new therapeutic strategies to target cancer-associated condensates (*Suzuki and Onimaru, 2022*; *Boija et al., 2021*). We recently found that TopBP1 condensation acts as a molecular switch to amplify ATR activity (*Frattini et al., 2021*), a critical pathway to tolerate the intrinsically high levels of lesions that block replication fork progression in cancer cells. Targeting TopBP1 assembly, rather than its degradation, offers an interesting strategy to selectively inhibit its role in the ATR/Chk1 signaling pathway activation, while preserving its other essential functions in replication and transcription that do not rely on its assembly.

Overall, this study brought the first insights into the rationale of targeting the condensation of TopBP1, a multifunctional protein that forms biomolecular condensates in response to DNA damage to inhibit the ATR/Chk1 signaling pathway and to potentiate conventional therapies.

# Materials and methods
## Cell culture

Flp-In 293 T-REx cells were obtained from Thermo Fisher and cultured in Dulbecco's Modified Eagle Medium (DMEM)-GlutaMAX medium (Merck-Sigma-Aldrich, #D5796) supplemented with 10% heat-inactivated fetal bovine serum (FBS). For the generation of stable cell lines that express TopBP1 fused to the photoreceptor cryptochrome 2 (Cry2) and mCherry (optoTopBP1), please refer to *Frattini et al., 2021*. The HCT116 human CRC cell line was obtained from Horizon (#HD-PAR-082). The human SW620, SW480, HT29, and murine CT26 CRC cell lines from ATCC (CT26.WT -CRL-2638, HCT116-CCL247, HT29-HTB-38, SW620-CCL-227, SW480-CCL228). The control human colon cell line CCD 841 CoN was also obtained from ATCC (CCD 841 CoN-CRL-1790). The SN-38-resistant clones HCT116-SN6 (low-level resistance) and HCT116-SN50 (high-level resistance) were generated in the laboratory as previously described (*Candeil et al., 2004*). Briefly, parental HCT116 cells were grown in the presence of 10 nmol/L or 15 nmol/L SN-38, respectively. SW620 shLuc and SW620 shGSK3-β cells were kindly provided by Maguy Del Rio. The description of their generation can be found in *Cherradi*

*et al., 2023*. All CRC cell lines were grown in and cultured in RPMI medium (Merck-Sigma-Aldrich, #R8758) supplemented with 10% heat-inactivated FBS. All cell lines were maintained at 37°C in saturated humidity with 5% $CO_2$. All cell lines were tested and authenticated by short-tandem repeat profiling (Eurofins Genomics). Cells were routinely tested for mycoplasma contamination.

### Drugs and antibodies

A personalized TargetMol library containing 4730 molecules was used for the high-throughput screening (see Screening section). SN-38 (active metabolite of irinotecan, Tocris #2684) was diluted to 10 mM in DMSO and stored at –20°C. 5-FU (Sigma, #F6627) was diluted to 10 mM in water and stored at room temperature (RT). AZD2858 (Euromedex, #AB-M2178) was diluted to 10 mM in DMSO and stored at –80°C. The drugs were diluted in the adapted medium and tested at the concentrations indicated in the figures and legends.

The following primary and secondary antibodies were used as indicated in *Supplementary file 6*.

### Screening

OptoTopBP1-expressing Flp-In 293 T-Rex cells were seeded in 384-well plates at a density of 10,000 cells/well in DMEM supplemented with 10% FBS and 2 μg/mL doxycycline and allowed to attach overnight. Then, cells were incubated with the TargetMol library compounds at the concentration of 10 μM for 2 hr. From the 10 mM stock solutions in DMSO, each compound was diluted to 2 mM (intermediate dilution) before addition to the wells by automated pipetting at a final concentration of 10 μM. After incubation, cells were exposed to cycling pulses (4 s 'ON', 10 s 'OFF') of 488 nm blue light for 5 min. Cells were fixed immediately in 4% paraformaldehyde (PFA) (Euromedex, #15710S) diluted in phosphate-buffered saline (PBS) for 15 min. Cells were rinsed with PBS twice and permeabilized in PBS/0.2% Triton X-100 for 10 min. Cells were rinsed in PBS twice, and nuclei were counterstained with 1 μg/mL Hoechst 33342 (Thermo Fisher Scientific, #62249). Imaging was performed with the Opera PHENIX system at the Montpellier Ressources Imagerie facility.

### Western blotting

Whole cell extracts were obtained by lysing cells in RIPA buffer (50 mM Tris, 150 mM NaCl, 1% NP-40, 1% deoxycholate, 0.1% SDS, pH 8) on ice for 30 min. After sonication (40% amplitude, 3 cycles of 3 s sonication and 3 s resting), the protein amount was quantified using the Pierce BCA Protein Assay Kit (Thermo Fisher #23225). Laemmli buffer was added, and protein extracts were boiled at 95°C for 5 min. 40 μg of protein samples were resolved on precast SDS-PAGE gels (4–15%, 7.5%, and 10%, Bio-Rad) and transferred to nitrocellulose membranes using the Bio-Rad Trans-Blot Turbo transfer device. Membranes were saturated in 5% non-fat milk diluted in TBS (20 mM Tris, 150 mM NaCl, pH 7.4)-0.1% Tween 20, and incubated with primary antibodies at 4°C overnight. The used antibodies were against Chk1 phosphorylated at S345 (Cell Signaling Technology, #2348), Chk1 (Santa Cruz Biotechnology, #sc8408), Chk2 phosphorylated at T68 (Cell Signaling Technology, #2661S), Chk2 (Milipore, #05-649), TopBP1 (Euromedex, #A300-111A or Santa Cruz Biotechnology, #sc-271043), ATM phosphorylated at S1981 (Rockland, #200-301-400), α-tubulin (Sigma, #T5168), RPA phosphorylated at S33 (Abcam, #ab2118877), RPA (Abcam, #AB2175), GSK-3β phosphorylated at S9 (Cell Signaling Technology, #9336), GSK-3β (Cell Signaling Technology, #9832), PARP1 (Santa Cruz Biotechnology, #sc-8007), cleaved caspase-3 (Cell Signaling Technology, #9661). For more information, please refer to *Supplementary file 6*. Membranes were then incubated with anti-mouse (Cell Signaling Technology, #7076S) or anti-rabbit HRP (Cell Signaling Technology, #7074S) secondary antibodies for 1 hr. Revelation was carried out with ECL Clarity (Bio-Rad, #170-5061) or ECL select (Bio-Rad, #1705062) according to the manufacturer's instructions.

### Immunofluorescence analysis

Cells were grown on coverslips and after incubation with the corresponding drugs (see above) for 2 hr, the soluble fraction of cells was removed during the pre-extraction step in Cytoskeleton buffer 'CSK' (PBS containing 0.2% Triton X-100) at 4°C for 60 s. Cells were then fixed with 4% PFA/PBS at RT for 20 min. Nonspecific epitopes were saturated with a blocking solution (5% BSA/PBS) at RT for 30 min. For immunostaining, anti-TopBP1 (Santa Cruz Biotechnology, #sc-271043, diluted at 1/100), anti-53BP1 (Cell Signaling Technology, #4937S) or anti-PML (Santa Cruz Biotechnology, #sc-966, diluted at

1/300) primary antibodies and the anti-mouse coupled to fluorochrome (Invitrogen, #A-11011, diluted at 1/500) secondary antibody were diluted in blocking solution and incubated for 1 hr or 45 min, respectively. Three 5 min washes between incubation steps were performed in PBS/0.1% Tween 20 at RT. Hoechst was used at 1 µg/mL for DNA staining, and coverslips were mounted on slides with the Prolong Gold antifade reagent (Invitrogen, #P36930). Images were captured using a 63X objective and a Zeiss AxioImager with ApoTome at the Montpellier Ressources Imagerie facility.

## TurboID assay: pull-down of biotinylated proteins

Flp-In 293 T-Rex cells that express optoTopBP1 and grown to 75% of confluence were incubated with 2 µg/mL of doxycycline for 16 hr. The next day, during the last 15 min of incubation with the indicated drugs, 500 mM of biotin was added. Cells were then washed with PBS and lysed in lysis buffer (50 mM Tris-HCl pH 7.5, 150 mM NaCl, 1 mM EDTA, 1 mM EGTA, 1% NP-40, 0.2% SDS, 0.5% sodium deoxycholate) supplemented with 1X complete protease inhibitor, 1X phosphatase inhibitor, and 250 U benzonase. Lysed cells were incubated on a rotating wheel at 4°C for 1 hr before sonication (40% amplitude, 3 cycles of 1 s sonication – 2 s resting) on ice. After centrifugation at 7750 rcf at 4°C for 30 min, cleared supernatants were transferred to new tubes, and the total protein concentration was determined using the Bradford protein assay (Bio-Rad). For each condition, 1 mg of proteins was incubated with 100 µL of streptavidin-agarose beads on a rotating wheel at 4°C for 3 hr. After centrifugation at 400 rcf for 1 min, beads were washed sequentially with 1 mL lysis buffer, 1 mL wash buffer 1 (2% SDS in $H_2O$), 1 mL wash buffer 2 (0.2% sodium deoxycholate, 1% Triton X-100, 500 mM NaCl, 1 mM EDTA, and 50 mM HEPES, pH 7.5), 1 mL wash buffer 3 (250 mM LiCl, 0.5% NP-40, 0.5% sodium deoxycholate, 1 mM EDTA, 500 mM NaCl, and 10 mM Tris pH 8) and 1 mL wash buffer 4 (50 mM Tris pH 7.5 and 50 mM NaCl). Bound proteins were eluted from the agarose beads with 80 µL of 2X Laemmli sample buffer and incubated at 95°C for 10 min. Western blot analysis was performed as previously described to analyze the self-proximity of optoTopBP1 molecules.

## X. laevis egg extract assay

Interphasic Low Speed Egg extracts (LSE) and demembranated sperm nuclei were prepared as previously described (Cohen et al., 1999). LSE were then clarified by centrifugation at 20,000 rpm in an SW55Ti rotor for 20 min. Chromatin preparation was performed as described before (Aze et al., 2016). Briefly, demembranated sperm nuclei were added at a final concentration of 4000 nuclei per microliter to LSE supplemented with energy regeneration mix (200 µg/mL creatine phosphokinase, 200 mM creatine phosphate, 20 mM ATP, 20 mM $MgCl_2$, 2 mM EGTA) and cycloheximide (250 µg/mL). When indicated, LSE were preincubated with camptothecin (CPT), AZD2858, or DMSO (as control) for 5 min. The mixtures were incubated at 23°C for the indicated times, then samples were diluted in EB buffer (100 mM KCl, 50 mM HEPES-KOH, 2.5 mM $MgCl_2$, supplemented with 0.25% NP40) and centrifuged over a 30% sucrose cushion. Purified chromatin pellets were washed once with EB buffer and recovered in Laemmli buffer for western blot analysis. Antibody references are provided in Supplementary file 6.

## Cell cycle assay

Cell cycle profiling using 5-bromo-2′-deoxyuridine (BrdU)/propidium iodide (PI) was performed as previously described (Egger et al., 2022). Briefly, cells were pulsed with 10 µM BrdU for 15 min and fixed in ice-cold 70% ethanol. Cells were digested in 30 mM HCl, 0.5 mg/mL pepsin (37°C, 20 min). DNA was denatured in 2N HCl (RT, 20 min), and BrdU was immunodetected using the anti-BrdU clone BU1/75 in PBS/2% goat serum/0.5% HEPES/0.5% Tween 20 and revealed with a goat anti-rat Alexa Fluor 488 secondary antibody. DNA was stained with 25 µg/mL PI in PBS, and cells were analyzed on a Gallios flow cytometer (Beckman Coulter) at the Montpellier Ressources Imagerie facility. 20,000 cells/ sample were analyzed using the Kaluza software.

## Detection of apoptotic cells with the sub-G1 assay

For visualization of sub-G1 apoptotic cells by flow cytometry, the 'SubG1 Analysis Using Propidium Iodide' protocol developed by the UCL – London's Global University was used. Briefly, cells were collected and fixed in ice-cold 70% ethanol, followed by two washes in phosphate-citrate buffer (192 mM $Na_2HPO_4$, 4 mM citric acid, pH 7.8). Ribonuclease A (100 µg/mL in PBS) digestion was

performed at 37°C for 30 min. DNA was stained with PI (50 µg/mL in PBS) for 15 min. Cells were analyzed on a Gallios flow cytometer (Beckman Coulter) at the Montpellier Ressources Imagerie facility. 20,000 cells/sample were analyzed with the Kaluza software. Debris (low FSC/SSC) was kept, and the sub-G1 fractions were calculated as the percentage of sub-G1 (<2N PI content) cells relative to the total number of events.

## Pulse field gel electrophoresis

Cells were collected and embedded in 0.5% agarose plugs (in PBS), at the exact concentration of $0.7 \times 10^6$ cells per plug. Plugs were incubated in lysis buffer (100 mM EDTA pH 8, 0.2% sodium deoxycholate, 1% sodium lauryl sarcosine, 1 mg/mL proteinase K) at 37°C for 48 hr. Plugs were washed three times in wash buffer (20 mM Tris pH 8, 50 mM EDTA pH 8) and inserted in 0.9% agarose gel prepared in 0.5X TBE (from a 5X stock PanReac-AppliChem #A4228). Chromosomes were separated by pulsed-field gel electrophoresis for 23 hr (Biometra Rotaphor 8 System, 23 hr; interval: 30-5 s log; angle: 120-110 linear; voltage: 180–120 V log, 13°C). Then, gels were stained with ethidium bromide (0.5 µg/mL) for analysis.

## 2D cell growth inhibition assay

Cell growth was evaluated using the sulforhodamine B (SRB) assay, as described by *Skehan et al., 1990*. Briefly, 500 cells/well were seeded in 96-well plates. After 24 hr, cells were incubated with the indicated drug concentrations for 96 hr. Cells were fixed in 10% trichloroacetic acid solution (Sigma, #T9159), rinsed in water, and stained with 0.4% SRB (Sigma, #S9012) in 1% acetic acid (Fluka, #33209). Plates were washed three times with 1% acetic acid and fixed SRB was dissolved in 10 mmol/L Tris-Base solution (Trizmabase Sigma, #T1503). The absorbance was read at 560 nm using a PHERAstar FS plate reader (BMG Labtech). Non-treated controls were used to normalize the cell growth inhibition values. The $IC_{50}$ of drugs was determined graphically using the cell growth inhibition curves.

## 3D spheroid assay

For spheroid generation, 100 µL/well of cell suspensions at optimized densities (50 cells/well) were dispensed in ultralow attachment 96-well round-bottomed plates (Corning B.V. Life Sciences, #7007) and cultured at 37°C, 5% $CO_2$, 95% humidity. Cells were first incubated with the indicated drugs at day 1 and then at day 4. At day 7, images were captured with the Celigo Imaging Cytometer (Nexcelom Bioscience) using the 'Tumorosphere' application. Cell viability was measured using a CellTiter-Glo Luminescent Cell Viability Assay (Promega, #G9683), according to the manufacturer's instructions. Luminescence was measured in the white 96-well plates using a PHERAstar FS plate reader (BMG LABTECH). The $IC_{50}$ was determined graphically from the cytotoxicity curves.

## Synergy matrix

The percentage of living cells after incubation with each drug alone or in combination was calculated and normalized to that of untreated cells. Then, using a script developed by *Tosi et al., 2018* in the 'R' software based on the effect of each molecule alone (Bliss and Lehàr equation), a synergy matrix was generated. This matrix associates a number to each drug combination: if positive (red), it indicates synergistic effects; if negative (green), it indicates antagonistic effects. Additive effects (~0) are displayed in black.

## In vitro cytotoxicity assays

Cells were seeded at 500 cells/well in black flat-bottom 96-well plates (Sarstedt, #83.3924), cultured for 24 hr, then incubated with AZD2858 and/or FOLFIRI for 96 hr. Then, PI (Sigma, #P4864) and Hoechst 33342 (Thermo Fisher Scientific, #62249) were added to the plates at the final concentrations of 1 and 5 µg/mL, respectively. Cells were incubated at 37°C for 30 min before counting the positive cells for each fluorescence signal using a Celigo Imaging Cytometer (Nexcelom Bioscience) and the 'Expression analysis – Cell Viability – Dead +Total' application. The number of living cells was then calculated by subtracting the number of dead, PI-positive cells from the number of total cells given by the Hoechst staining. The percentages of live and dead cells were plotted, and cytotoxic and cytostatic profiles were determined, as previously described by *Vezzio-Vié et al., 2022*.

## Celigo imaging cytometer-based immunofluorescence analysis

HCT116 cells were seeded at 1200 cells per well for 48 or 24 hr in black 384-well plates (Greiner, #781091) and incubated with increasing concentrations of AZD2858 and/or FOLFIRI (diluted to 1/2, corresponding to 50 nM of SN-38 and 6 μM of 5-FU) for 2 and 20 hr, respectively. Then, cells were fixed in 4% PFA, permeabilized in PBS/0.1% Triton X-100 (Sigma), and saturated in 3% BSA/PBS. The primary antibodies anti-Chk1 phosphorylated at S345 (Cell Signaling Technology, #2348L), anti-Chk2 phosphorylated at T68 (Cell Signaling Technology, #2661S), and anti-ATM phosphorylated at S1981 (Santa Cruz Biotechnology, #sc-47739) were diluted in 1% BSA/PBS and cells were incubated at 4°C under stirring overnight. Cells were washed three times with 0.05% Tween 20/PBS (Tween20 Polysorbate Technical VWR CHEMICALS). The Alexa Fluor 568 goat anti-rabbit or anti-mouse IgG (H+L) (Thermo Fisher Scientific, #A11011 or #A11004, respectively) secondary antibody was added at RT for 45 min. The anti-γH2AX (phosphorylated at S139) (Abcam, #195189) antibody was diluted (1/1000) in 1% BSA/PBS and added to the cells at RT for 45 min. Cells were washed three times and 1 μg/mL of Hoechst 33342 in PBS was used to counterstain nuclei (37°C, 30 min). Fluorescence signals were acquired using the 'Expression analysis – Target 1+Mask' application on the Celigo Imaging Cytometer (Nexcelom). The percentage of positive cells, the mean signal intensity per well, and the total cell count were calculated using the Nexcelom Bioscience Celigo Satellite software.

## DNA fiber assay

$1 \times 10^6$ HCT116 cells were seeded in six-well plates. Cells were allowed to attach overnight and then sequentially pulsed with 20 μM 5-iodo-2'-deoxyuridine (IdU) (20 min) and 200 μM 5-chloro-2'-deoxyuridine (CldU) (20 min). Cells were harvested, washed, and diluted in ice-cold PBS to $0.5–1 \times 10^6$ cells/mL. 2 μL of cell suspension was pipetted on the edges of SuperFrost microscopy slides. After drying at RT for 3–5 min, 7 μL of DNA Spreading Buffer (200 mM Tris-HCl pH 7.5, 50 mM EDTA, 0.5% SDS) was mixed with the cell-containing drops. After drying at RT for 2–5 min, slides were manually tilted (15–30°) to spread the DNA fibers that were then fixed in a 3:1 methanol/acetic acid solution at RT for 10 min. Slides were washed in $H_2O$ and DNA was denatured in 2.5 M HCl for 1 hr. Slides were washed and blocked in PBS/1% BSA/0.1% Tween 20 at RT for 1 hr. Then, slides were incubated with a mix of anti-IdU (BD Biosciences, Cat# 347580; RRID:AB_10015219) and anti-CldU (Bio-Rad, Cat# OBT-0030, RRID:AB_2314029) antibodies, both diluted (1/25) in PBS/0.1% Tween 20, at 37°C in saturated humidity for 45 min. Slides were washed in PBS/0.1% Tween 20 five times (2 min/each) and incubated with a mix of secondary antibodies, diluted 1/50 in PBS/0.1% Tween 20, at 37°C for 30 min. Slides were rinsed in PBS/0.1% Tween 20 five times (2 min/each), dried, and mounted with $24 \times 50$ mm² coverslips in 30 μL of ProLong Gold Antifade. Twenty representative pictures for each condition were saved on a Zeiss Axio Imager (40X objective). The CldU tract lengths of individual fibers consisting of a sequence of IdU and CldU tracts were measured using the open-source FiberQ software (*Li et al., 2022*). Experiments were repeated three times, and the pooled data from three replicates were plotted using the online tool SuperPlotsOfData (*Goedhart, 2021*). The non-parametric unpaired Mann-Whitney test was used to determine the significance of the results.

## Software tools

Immunofluorescence data were acquired with a Zeiss AXIOIMAGER microscope (Zeiss) and the Zen toolkit connect software. For foci quantification, the pipelines generated on CellProfiler 4.2.1 and described in *Egger et al., 2024*, were used. For western blot images acquisition, ChemiDoc Imaging System (Bio-Rad) and ImageLab 6.1 were used. The RStudio software was used to generate the synergy and viability matrices with a predefined script. GraphPad version 8 was used for graphical representation and statistical analyses. R was used for statistical analyses. The DNA fibers were quantified using open-source FiberQ software. For analysis of TopBP1 foci in parental and SN-38-resistant HCT116 cells, and for DNA fiber assay, biological replicates were pooled and plotted using SuperPlotsOfData (https://huygens.science.uva.nl/SuperPlotsOfData/) (*Whelan and Rothenberg, 2021*). Data distribution was displayed as half-violin plots on the right-hand side of the dot plots to help data visualization.

## Statistical analyses

For the immunofluorescence quantification of foci (TopBP1 [*Figure 2A*] and 53BP1 [*Figure 2—figure supplement 1A*]), all conditions were first compared using one-way ANOVA and confirmed using

non-parametric unpaired Mann-Whitney tests on GraphPad Prism version 10.2.1 (the non-normal distributions of foci limited the precision of the ANOVA). SN-38 vs. SN-38+AZD2858 (Mann-Whitney); TopBP1: p-value<0.0001; 53BP1: p-value<0.0001. In addition, we conducted in-depth statistical analyses on R to complete the results obtained with GraphPad. Specifically, for TopBP1 foci, an ANOVA of a generalized linear model (modeling TopBP1 foci number with a Poisson distribution) was performed to confirm this. The ANOVA test shows that SN38 and AZD2858 treatments exhibit significantly negative interaction on TopBP1 foci number (SN38 treatment p-value<2.2e-16; AZD2858 treatment p-value<2.2e-16; SN38:AZD2858 interaction p-value<2.2e-16). For PML foci *Figure 2— figure supplement 1B*, all conditions were first compared using Kruskal-Wallis on GraphPad Prism and p-value<0.0001 was obtained. Then, Dunn's multiple comparison test was conducted to specifically compare the Arsenic condition with the following treatments: untreated cells, AZD2858, SN-38, and SN-38+AZD2858. In each comparison, the p-value was <0.0001.

For DNA fiber analyses, groups were first compared using one-way ANOVA and confirmed using unpaired t-tests on GraphPad Prism version 10.2.1 (the non-normal distributions of CldU tracts length limited the precision of the ANOVA analyses, *Figure 3B*). In addition, we conducted in-depth statistical analyses on R to complete the results obtained with GraphPad: ANOVA of a generalized linear model (modeling replication tract length with a gamma distribution) shows that both drugs and their interaction have a significant effect on tract length (p-value for SN38 treatment: <2.2e-16; for AZD2858 treatment: 0.01129; for their interaction: 2.651e-06). We did not detect a significant effect of biological replicate identity (p-value=0.26274).

For cytometry analyses, linear regression analysis was performed on R (*Figure 4B*). For *Figure 3*, *Figure 3—figure supplement 1*, we performed an ANOVA to evaluate the effect of chase duration (6 hr vs. 12 hr) and treatment (untreated vs. AZD2858 vs. SN38 vs. AZD2858+SN38) on the proportion of cells in S-phase (endpoint, BrdU-positive cells), S-phase (pulse-chase, BrdU-negative cells, between 2N and 4N, green squares) and G2 phase (pulse-chase, BrdU-positive cells, 4N, red squares). For S-phase analysis, both treatment duration (6 hr vs. 12 hr) and nature of treatment (untreated, AZD2858-treated, SN38-treated or (AZD2858+SN38)-treated) have a significant effect on the proportion of cells in S-phase in the endpoint experiment (p=0.00308 for treatment duration, p=2.14e-06 for nature of treatment). Interaction between treatment nature and duration did not appear significant (p=0.05942): AZD2858-treated and untreated cells tend to have higher S-phase cell counts than SN38-treated and (AZD2858+SN38)-treated cells, regardless of treatment duration; and increasing treatment duration (from 6 to 12 hr) decreases S-phase cell count, regardless of treatment nature. For S-entry analysis, both treatment duration (6 hr vs. 12 hr) and nature of treatment (untreated, AZD2858-treated, SN38-treated, or (AZD2858+SN38)-treated) have a significant effect on the proportion of cells in SE phase in the pulse-chase experiment (p=0.0173 for treatment duration, p=4.03e-06 for nature of treatment). Interaction between treatment nature and duration also has a significant effect (p=0.00865): AZD2858-treated and untreated cells tend to have higher SE cell counts than SN38-treated and (AZD2858+SN38)-treated cells, but that difference is larger after a 6 hr treatment than a 12 hr treatment. For G2 phase analysis, treatment duration (6 hr vs. 12 hr) has a significant effect on the proportion of cells in G2 phase in the pulse-chase experiment (p=0.000363). The nature of the administered treatment significantly alters the effect of duration (interaction between treatment nature and treatment duration: p=0.000340), with SN38-treated and (AZD2858+SN38)-treated cells exhibiting similar proportion of G2 cells after 6 and 12 hr of treatment, while untreated and AZD2858-treated cells undergo a drop in G2 cell count between 6 and 12 hr of treatment. We confirmed these analyses with conventional t-test on GraphPad Prism: S-phase (6 hr, 12 hr) p-value<0.01; S-entry (12 hr) p-value<0.01; G2-phase (12 hr) p=0.0654 (>0.02, ns: nonsignificant).

In all cases, significance thresholds were as follows: ns: nonsignificant; *: p<0.05; **: p<0.01; ***: p<0.001; ****: p<0.0001.

## Inclusion and diversity

We support inclusive, diverse, and equitable conduct of research.

# Additional information

## Funding

| Funder | Grant reference number | Author |
|---|---|---|
| Institut National du Cancer | PLBIO2021 | Céline Gongora |
| Agence Nationale de la Recherche | AAPG2021 | Angelos Constantinou |
| Agence Nationale de la Recherche | ANR-10-INBS-04 | Benoit Bordignon |
| Fondation MSD Avenir | | Angelos Constantinou |

The funders had no role in study design, data collection and interpretation, or the decision to submit the work for publication.

## Author contributions

Laura Morano, Conceptualization, Data curation, Formal analysis, Investigation, Methodology, Writing – original draft; Nadia Vezzio-Vié, Formal analysis, Supervision, Investigation, Methodology; Adam Aissanou, Formal analysis, Investigation, Methodology, Writing – original draft; Tom Egger, Formal analysis, Investigation, Methodology, Writing – original draft, Writing – review and editing; Antoine Aze, Formal analysis, Methodology, Writing – original draft; Solène Fiachetti, Louis-Antoine Milazzo, Formal analysis, Methodology; Benoit Bordignon, Conceptualization, Formal analysis, Investigation, Methodology, Writing – original draft; Cedric Hassen-khodja, Formal analysis, Investigation, Methodology; Hervé Seitz, Data curation, Software, Formal analysis, Investigation, Writing – original draft; Véronique Garambois, Methodology; Laurent Chaloin, Formal analysis; Nathalie Bonnefoy, Funding acquisition; Céline Gongora, Conceptualization, Formal analysis, Supervision, Funding acquisition, Validation, Investigation, Methodology, Writing – original draft, Project administration, Writing – review and editing; Angelos Constantinou, Conceptualization, Supervision, Funding acquisition, Investigation, Writing – original draft, Project administration; Jihane Basbous, Conceptualization, Supervision, Validation, Investigation, Writing – original draft, Project administration, Writing – review and editing

## Author ORCIDs

Hervé Seitz  https://orcid.org/0000-0001-8172-5393
Céline Gongora  https://orcid.org/0000-0001-9034-4031
Angelos Constantinou  https://orcid.org/0000-0002-2994-8140
Jihane Basbous  https://orcid.org/0000-0002-3943-627X

Reviewer #1 (Public review): https://doi.org/10.7554/eLife.106196.3.sa1
Reviewer #2 (Public review): https://doi.org/10.7554/eLife.106196.3.sa2
Reviewer #3 (Public review): https://doi.org/10.7554/eLife.106196.3.sa3
Author response https://doi.org/10.7554/eLife.106196.3.sa4

# Additional files

## Supplementary files

Supplementary file 1. GSK-3 inhibitors in the TargetMol library. The specificity column indicates the $IC_{50}$ values available in the literature corresponding to 50% of the maximal concentration needed to inhibit the GSK-3 target (except for GSK3i XIII, where only the inhibition constant, Ki, value was available). The GSK-3β specificity column indicates the specificity toward this isoform, if available. If specificity was determined, but the GSK-3β isoform was not clearly specified, a question mark is used to indicate this uncertainty (for VP3.15 dihydrobromide and AT 7519 hydrochloride salt). ++++: <1 nM; +++: between 1 and 10 nM; ++: between 10 and 40 nM; +: >40 nM. Molecules that inhibited light-induced optoTopBP1 foci in the present screen are highlighted in red, and the z-score is indicated. The SN-38-induced Chk1 phosphorylation (pChk1) inhibition column indicates whether the potential GSK-3 inhibitors from the initial screen inhibit SN-38-induced Chk1 phosphorylation at S345 in HCT116 cells. N/A: not available.

Supplementary file 2. Three biological replicates' results of flow cytometry experiments of the 2 hr condition (endpoint), including the representative one shown in *Figure 3A*. AZD2858: 100 nM. SN-38: 300 nM.

Supplementary file 3. Three biological replicates' results of flow cytometry experiments of the 6 and 12 hr condition (endpoint and pulse chase), including the representative one shown in *Figure 3C*, *Figure 3—figure supplement 1C and D*. AZD2858: 100 nM. SN-38: 300 nM.

Supplementary file 4. Results of the linear regression analysis (*Figure 4A and B*). NT: non-treated. A: AZD2858. S: SN-38. F: FOLFIRI (5-FU+SN-38). The p-values of interest from *Figure 4B* are indicated in red.

Supplementary file 5. FOLFIRI concentration ranges used for the synergy matrix assays. All the dilutions are based on the dilution 1/1 corresponding to 12 µM 5-fluorouracil (5-FU) and 100 nM SN-38.

Supplementary file 6. References of the antibodies used in the study. WB, western blotting, IF, immunofluorescence analyses.

MDAR checklist

## Data availability

Detailed R scripts and data files for the analysis of data shown in Figures 2A, C, 3B, 4B, Figure 1–figure supplement 1, Figure 2–figure supplement 2 and Figure 3–figure supplement 1 are available at https://github.com/HKeyHKey/Morano_2025 (copy archived at *Seitz, 2025*).

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
