## [Editor Report · eLife Assessment]

This **valuable** study reveals that the GSK-3 inhibitor AZD2858 inhibits the formation of TOPBP1 condensates and hence DNA damage responses in colorectal cancer cells. The evidence supporting the claims of the authors is **convincing**, although uncovering how this drug blocks bio-condensate formation would have strengthened the study. The work will be of interest to cancer researchers searching for synergistic drug combination strategies.

[Editors' note: this paper was reviewed by Review Commons.]

---

## [Referee Report · Reviewer #1 (Public review)]

Summary:

Laura Morano and colleagues have performed a screen to identify compounds that interfere with the formation of TopBP1 condensates. TopBP1 plays a crucial role in the DNA damage response, and specifically the activation of ATR. They found that the GSK-3b inhibitor AZD2858 reduced the formation of TopBP1 condensates and activation of ATR and its downstream target CHK1 in colorectal cancer cell lines treated with the clinically relevant irinotecan active metabolite SN-38. This inhibition of TopBP1 condensates by AZD2858 was independent from its effect on GSK-3b enzymatic activity. Mechanistically, they show that AZD2858 thus can interfere with intra-S-phase checkpoint signaling, resulting in enhanced cytostatic and cytotoxic effects of SN-38 (or SN-38+Fluoracil aka FOLFIRI) in vitro in colorectal carcinoma cell lines.

Comments on latest version:

The requested plots are in figure S7 of the latest manuscript version, and look convincing. My last point is now adequately addressed.

---

## [Referee Report · Reviewer #2 (Public review)]

Summary:

In 2021 (PMID: 33503405) and 2024 (PMID: 38578830) Constantinou and colleagues published two elegant papers in which they demonstrated that the Topbp1 checkpoint adaptor protein could assemble into mesoscale phase-separated condensates that were essential to amplify activation of the PIKK, ATR, and its downstream effector kinase, Chk1, during DNA damage signalling. A key tool that made these studies possible was the use of a chimeric Topbp1 protein bearing a cryptochrome domain, Cry2, which triggered condensation of the chimeric Topbp1 protein, and thus activation of ATR and Chk1, in response to irradiation with blue light without the myriad complications associated with actually exposing cells to DNA damage.

In this current report Morano and co-workers utilise the same optogenetic Topbp1 system to investigate a different question, namely whether Topbp1 phase-condensation can be inhibited pharmacologically to manipulate downstream ATR-Chk1 signalling. This is of interest, as the therapeutic potential of the ATR-Chk1 pathway is an area of active investigation, albeit generally using more conventional kinase inhibitor approaches.

The starting point is a high throughput screen of 4730 existing or candidate small molecule anti-cancer drugs for compounds capable of inhibiting the condensation of the Topbp1-Cry2-mCherry reporter molecule in vivo. A surprisingly large number of putative hits (>300) were recorded, from which 131 of the most potent were selected for secondary screening using activation of Chk1 in response to DNA damage induced by SN-38, a topoisomerase inhibitor, as a surrogate marker for Topbp1 condensation. From this the 10 most potent compounds were tested for interactions with a clinically used combination of SN-38 and 5-FU (FOLFIRI) in terms of cytotoxicity in HCT116 cells. The compound that synergised most potently with FOLFIRI, the GSK3-beta inhibitor drug AZD2858, was selected for all subsequent experiments.

AZD2858 is shown to suppress the formation of Topbp1 (endogenous) condensates in cells exposed to SN-38, and to inhibit activation of Chk1 without interfering with activation of ATM or other endpoints of damage signalling such as formation of gamma-H2AX or activation of Chk2 (generally considered to be downstream of ATM). AZD2858 therefore seems to selectively inhibit the Topbp1-ATR-Chk1 pathway without interfering with parallel branches of the DNA damage signalling system, consistent with Topbp1 condensation being the primary target. Importantly, neither siRNA depletion of GSK3-beta, or other GSK3-beta inhibitors were able to recapitulate this effect, suggesting it was a specific non-canonical effect of AZD2858 and not a consequence of GSK3-beta inhibition per se.

To understand the basis for synergism between AZD2858 and SN-38 in terms of cell killing, the effect of AZD2858 on the replication checkpoint was assessed. This is a response, mediated via ATR-Chk1, that modulates replication origin firing and fork progression in S-phase cell under conditions of DNA damage or when replication is impeded. SN-38 treatment of HCT116 cells markedly suppresses DNA replication, however this was partially reversed by co-treatment with AZD2858, consistent with the failure to activate ATR-Chk1 conferring a defect in replication checkpoint function.

Figures 4 and 5 demonstrate that AZD2858 can markedly enhance the cytotoxic and cytostatic effects of SN-38 and FOLFIRI through a combination of increased apoptosis and growth arrest according to dosage and treatment conditions. Figure 6 extends this analysis to cells cultured as spheroids, sometimes considered to better represent tumor responses compared to single cell cultures.

Significance:

Liquid phase separation of protein complexes is increasingly recognised as a fundamental mechanism in signal transduction and other cellular processes. One recent and important example was that of Topbp1, whose condensation in response to DNA damage is required for efficient activation of the ATR-Chk1 pathway. The current study asks a related but distinct question; can protein condensation be targeted by drugs to manipulate signalling pathways which in the main rely on protein kinase cascades?

Here, the authors identify an inhibitor of GSK3-beta as a novel inhibitor of DNA damage-induced Topbp1 condensation and thus of ATR-Chk1 signalling.

This work will be of interest to researchers in the fields of DNA damage signalling, biophysics of protein condensation, and cancer chemotherapy.

Comments on latest version:

Having read the revised manuscript and rebuttal I am satisfied that the authors have resolved my various original concerns through a combination of clarification/ explanation and textual changes necessary to make the description of certain data precise. My impression is that they have also largely or completely satisfied the concerns of the other reviewers, with the possible exception of reviewer 1's point about the relative toxicity of AZD and FOLFIRI in colorectal cancer cell lines versus the untransformed CCD841 cell line. This is of course an important point with respect to the possible practical application of this combination for cancer therapy, however this seems somewhat subsidiary to the main novelty and significance of the findings, which are that protein liquid phase separation/ condensation can be manipulated pharmacologically to modify signal transduction processes and that existing drugs can be re-purposed to this end.

---

## [Referee Report · Reviewer #3 (Public review)]

Summary:

The authors have extended their previous research to develop TOPBP1 as a potential drug target for colorectal cancer by inhibiting its condensation. Utilizing an optogenetic approach, they identified the small molecule AZD2858, which inhibits TOPBP1 condensation and works synergistically with first-line chemotherapy to suppress colorectal cancer cell growth. The authors investigated the mechanism and discovered that disrupting TOPBP1 assembly inhibits the ATR/Chk1 signaling pathway, leading to increased DNA damage and apoptosis, even in drug-resistant colorectal cancer cell lines.

Comments on latest version:

This reviewer does not have further comments to the paper.

---

## [Author Response]

The following is the authors’ response to the original reviews

**Reviewer #1:**
Comments on revised version:I have reviewed the revised manuscript and read the rebuttal. The authors have carefully addressed my concerns. There is however one point that needs further work:This follows up on my major point #1 in my initial review. I had I asked the authors to demonstrate that FOLFIRI + AZD are less toxic to untransformed colorectal cells than colorectal cancer cell lines. It is good to see that the authors took my advice and show effects of the drug treatments on the untransformed colorectal cell line CCD841. It seems to be less sensitive to AZD and FOLFIRI in the figure in the rebuttal. What surprises me is that I cannot find these new figures anywhere in the revised manuscript. Also, the data seem preliminary, because I do not see any standard errors in the graphs, and I cannot find a description of the time of drug incubation. I ask the authors to make sure that the CCD841 data are reproducible, and make sure they incorporate the data in the revised manuscript.

We thank the reviewer for this insightful comment. In the initial revised version of the manuscript, we did not include results from the untransformed colorectal cell line CCD841, as those experiments had only been performed once and were considered preliminary. However, we fully agree with the reviewer on the importance of including these data.

To address this, we have repeated the experiments in CCD841 cells to ensure reproducibility. We now report the results from three independent experiments testing the combination of AZD2858 and FOLFIRI on healthy epithelial colon cells. These results are shown in Supplementary Figure S7, where blue matrices represent cell viability and black matrices reflect the level of synergy between AZD2858 and FOLFIRI.

Our results confirm that, individually, each drug has little to no effect on healthy cells, and no consistent synergistic interaction was observed, except in Experiment 1, which could not be reproduced. Importantly, the drug concentrations used were identical to those applied in the cancer cell experiments, allowing for direct comparison between normal and malignant cell responses.

**Reviewer #2:**
Comments on latest version:Morano et al. have revised their manuscript in response to the points raised by reviewer #3 as follows.(1) Fig. 2E: Correcting the previously erroneous labelling of this Fig. makes it match the textual description.(2) Figs 3A and B: The revised textual description of the flow cytometry BrdU incorporation is now precise.(3) Fig. 3E: Removing the suspect WB images is a pragmatic decision that does not significantly affect the overall conclusions of the paper.(4) Fig. 3D: Despite its puzzling appearance this data is now described accurately in the text as "DSBs remained elevated after the combined treatment" rather than "increased after the combined treatment. A more convincing increase in the presumed damaged DNA band is evident in Fig. 4D when AZD2858 is combined with a much lower concentration of SN38 (1.5nM) which could mean that the concentration used in Fig. 3D (300nM) induced maximal damage that could not be further enhanced.

We thank the reviewer for their thoughtful comments and constructive feedback, which have helped us improve the clarity and rigor of the manuscript.

**Reviewer #3:**
Comments on latest version:The authors have addressed most of the concerns that I raised in the first round of revision and I have no further questions. I appreciate the authors's efforts in carrying out an preliminary in vivo work, although as the authors pointed out the compound seems to be not effective in vivo. Future work is desired to address this to clarify the significance of the work.

We thank the reviewer for acknowledging our efforts in addressing the previous concerns. We also appreciate the recognition of our preliminary in vivo work. While these results suggest limited in vivo efficacy of the compound at this stage, we agree that additional studies will be necessary to fully evaluate its therapeutic relevance. We consider this an important next step and are committed to pursuing it in future work.